# C-Terminal 1-Aminoethyltetrazole-Containing Oligopeptides as Novel Alanine Racemase Inhibitors

**DOI:** 10.3390/molecules25061315

**Published:** 2020-03-13

**Authors:** Laszlo A. Kondacs, Sylvain Orenga, Rosaleen J. Anderson, Emma C.L. Marrs, John D. Perry, Mark Gray

**Affiliations:** 1Sunderland Pharmacy School, University of Sunderland, Sunderland SR1 3SD, UK; laszlo.kondacs@kcl.ac.uk (L.A.K.); roz.anderson@sunderland.ac.uk (R.J.A.); 2Research & Development Microbiology, bioMérieux, 38390 La Balme-les-Grottes, France; sylvain.orenga@biomerieux.com; 3Department of Microbiology, Freeman Hospital, Newcastle upon Tyne NE7 7DN, UK; e.marrs@nhs.net (E.C.L.M.); john.perry@nuth.nhs.uk (J.D.P.)

**Keywords:** alanine racemase, 1-aminoethyltetrazole, antibacterial agent, selective inhibition, solid phase peptide synthesis

## Abstract

In clinical culture media inoculated with patient samples, selective inhibition of commensal bacteria is essential for accurate diagnosis and effective treatment, as they can mask the presence of pathogenic bacteria. The alanine analogue, 1-aminoethyltetrazole was investigated as a potential alanine racemase inhibitor. For effective uptake and enhanced and selective antibacterial activity, a library of C-terminal 1-aminoethyltetrazole containing di- and oligopeptides were synthesized by solid phase peptide coupling techniques. The investigation of the antimicrobial activity of the synthesised compounds identified several clinically applicable selective inhibitors. These enabled differentiation between the closely related bacteria, *Salmonella* and *Escherichia coli*, which can be difficult to discriminate between in a clinical setting. In addition, differentiation between enterococci and other Gram-positive cocci was also seen.

## 1. Introduction

In order to treat bacterial infections effectively, and to avoid promoting the evolution of bacterial resistance against antibacterial drugs, the correct agents must be used. Therefore, rapid and accurate detection and identification of pathogenic strains are essential. Chromogenic and fluorogenic culture media are used routinely in the clinical environment for this purpose [1]. A limitation of such media occurs due to the presence of additional non-pathogenic bacteria which are commonly found in clinical samples, the presence of which could produce false signals. One way to overcome this problem is to add selective antibacterial agents to the medium that can specifically prevent the growth of irrelevant strains, but allow the pathogen under investigation to grow [2].

Our target was the widely-studied enzyme [3] alanine racemase (AlaR). This enzyme plays a key role in the biosynthesis of bacterial peptidoglycan through its activity in the racemisation of l-alanine to d-alanine in bacteria. In the past, several alanine racemase inhibitors have been reported [3,4], and some have been applied as antimicrobial agents. Examples of these include l,l-alafosfalin (**1**), the more active prodrug form of the direct AlaR inhibitor l-fosfalin (**2**). Figure 1 shows these known alanine racemase inhibitors, which are structurally similar to the natural substrate.

While l-fosfalin (**2**) is a good AlaR inhibitor in isolated enzyme studies [5], it does not produce significant antimicrobial effects when applied to intact bacteria. This is thought to be because it passes through the cell membrane in insufficient quantities to harm the bacteria. In turn, this is due to the highly ionised and polar character of this molecule, precluding passive diffusion. The more effective compound alafosfalin (**1**) contains two parts, the AlaR inhibitor fosfalin and a carrier alanine moiety. Despite also being highly polar, this molecule effectively inhibits the growth of many intact bacterial strains. This is because, unlike fosfalin itself, it is able to conscript the dipeptide permease system to cross the bacterial membrane and, after hydrolysis by an intracellular alanine aminopeptidase, it liberates fosfalin (**2**) as the active agent [6].

The cost of producing enantiomerically pure l-fosfalin is high and the antibacterial selectivity of its derivatives is limited [7]. Due to these factors, there is scope for the development of a new series of AlaR inhibitors. The tetrazole moiety has been utilised as a bioisostere for carboxylic acids in drugs such as Losartan, where problems occurred with membrane permeability earlier in the development cycle [8]. Thus, to see if a tetrazole bioisoteric replacement would produce interesting activity against AlaR in intact bacteria, l-1-aminoethyltetrazole **4a**-**L** was prepared and studied as an alanine (**3**) bioisostere (Figure 2). In addition, oligopeptide derivatives containing a C-terminal l-1-aminoethyltetrazole were synthesised from this initial material.

## 2. Results and Discussion

### 2.1. Chemistry

The synthesis of the key molecule l-1-aminoethyltetrazole **4a**-**L** has been published by Bavetsias [9]. This synthetic route was followed with minor modifications, producing **4a**-**L** in good overall yield (71%) (Scheme 1).

Cbz-L-Alanine **5a**-**L** was methylated with methyl iodide in the presence of Cs_2_CO_3_ in DMF. The resulting methyl ester **6a**-**L** was transformed to the analogous amide **7a**-**L** in the presence of 7 M ammonia in methanol by heating at 50 °C in a sealed tube. In the following step, the amide **7a**-**L** was converted to nitrile **8a**-**L** using tosyl chloride in pyridine and DCM. Cbz-l-1-aminoethyltetrazole **9a**-**L** was produced by a cycloaddition of the nitrile moiety with NaN_3_ in the presence of NH_4_Cl in DMF. l-1-Aminoethyltetrazole **4a**-**L** was obtained after removing the Cbz-protecting group by Pd-catalysed hydrogenation. The d-1-aminoethyltetrazole **4a**-**D** and l-1-aminopropyltetrazole **4b**-**L** were synthesized by the same methodology. l-1-Aminopropyltetrazole **5b**-**L** is a homologue that enabled a preliminary exploration of the steric requirements for tetrazole based inhibitors of AlaR. The justification for this is that (1-amino-2-propenyl)phosphonic acid is an inhibitor of AlaR, [10] which is the comparable homologue of the AlaR inhibitor fosfalin **2**.

A series of di- and oligopeptides containing C-terminal l-1-aminoethyltetrazoles were prepared by a solid phase synthetic approach. The tetrazole moiety has been successfully attached to and removed from different resins by several research groups [11,12,13,14], although to the best of our knowledge its application in peptide chemistry has never been reported. After careful consideration, 2-chlorotrityl chloride resin and the matching Fmoc approach were employed due to the mild cleavage conditions associated with this resin.

Fmoc-l-1-aminoethyltetrazole **10a**-**L** was prepared by protecting l-1-aminoethyltetrazole **4a**-**L** with Fmoc chloride in the presence of Na_2_CO_3_ [15], and then attached to the 2-chlorotrityl chloride resin (Scheme 2) [16]. In our optimised method, the resin was not loaded with an excess amount of the C-terminal unit, which in our case was Fmoc-1-aminoethyltetrazole **10**, as is commonly encountered in general practice. The synthetic procedure with fully loaded resin resulted in the isolation of Fmoc peptides as minor impurities at the end, resulting in the requirement for further purification steps. Instead, when loading with a theoretically equivalent ratio, reasonable yields were obtained after cleavage of the peptides (50–70%) free from the Fmoc peptide impurities that were encountered with a fully saturated resin, Table 1. The resin bound d-isomer **4a**-**D** and l-1-aminopropyltetrazole **4b**-**L** were prepared by the same method. 

A series of dipeptide derivatives of 1-aminoalkyltetrazole **14a,b,d**–**j** were synthesized using the same solid phase synthesis approach. First, the Fmoc protection of resin-attached Fmoc-l-1-aminoalkyltetrazole **11a,b** was removed with 25% piperidine solution in DMF. The free amino group was coupled with the appropriate Cbz protected amino acid using HBTU and DIPEA in DMF. The resin-bound dipeptides **12a,b,d**–**j** were cleaved from the resin by a TFA/DCM/triisopropylsilane (TIS) mixture in the ratio 5:95:1 eq (equivalent with the theoretical loading of the resin) and resulted in the protected dipeptides **13a,b,d**–**j** in good to excellent yields (Table 1). The Cbz-protecting group was removed by Pd-catalysed hydrogenation, Scheme 3; the yield of these reactions depended on the purity of the protected starting compound. 

A tripeptide analogue, l-alanyl-l-alanyl-l-1-aminoethyltetrazole **14-c-LLL** was accessed by coupling the attached l-1-aminoethyltetrazole **15-L** with an Fmoc-l-alanine, and then with a Cbz-l-alanine. Cbz-l-alanyl-l-alanyl-l-1-aminoethyltetrazole **13c-LLL** was cleaved from the resin with a mixture of 5% TFA, 95% DCM, 1 eq TIS. Subsequent Cbz deprotection by catalytic hydrogenation resulted in l-alanyl-l-alanyl-l-1-aminoethyltetrazole **14c-LLL** (Scheme 4).

A series of di-, tri- and tetra-peptidyl derivatives of N-terminal succinyl-, C-terminal l-1-aminoethyltetrazole **16-LL**, **17-LLL**, **18-LLLL** were synthesised by the same strategy. After the introduction of the appropriate number of alanine residues (1–3), the peptides were furnished with a succinyl group using succinic anhydride and DIPEA in DMF. These substrates were intended to target bacterial elastase enzymes in order to achieve a different pattern of selective inhibition. The rationale behind this lay in the knowledge that succinyl-tripeptides were cleaved from chromogenic peptidase substrates by *Pseudomonas aeruginosa* in a previous study [17].

AlthougH-N-Cbz protected peptides were successfully cleaved from the resin by a mixture of 5% TFA, 95% DCM, 1 eq TIS, with good solubility in the cleavage medium, the succinyl peptides **16-LL**, **17-LLL**, **18-LLLL** were not sufficiently soluble in this organic solvent mixture. Due to this they were instead cleaved from the resin by a better solubilising mixture of 95% TFA, 2.5% H2O, 2.5% TIS. This method was both effective and convenient, due to the lack of a need for a N-terminal protecting group, Scheme 5.

### 2.2. Microbiological Evaluation

All of the synthesised 1-aminoethyltetrazole derivatives were tested against a panel of clinically relevant pathogenic bacteria to investigate their antibacterial properties. The minimum inhibitory concentration (MIC) tests were carried out in antagonist-free, blood supplemented agar media containing the appropriate 1-aminoethyltetrazole compound in different concentrations (1-128 mg/L). l-Alanyl-l-alanyl-l-1-aminoethyltetrazole **14c-LLL** and l-alanyl-l-1-aminoethyltetrazole **14a-LL** have been synthesized and tested against a selected group of bacteria in the past: *Escherichia coli*, *Staphylococcus aureus* and *Salmonella enterica* serovar Dublin. In this previous work they were reported to be inactive, which contrasts starkly with the data we report below. However, these differences in results may be attributed to differences in the culture medium that was used, such as peptone content, the specific bacterial strains studied or even differences in the inoculum [18,19]. Our more extensive investigation focused upon clinically relevant strains of pathogenic bacteria, and used protocols that have now been established as clinical standard over many years.

l-1-Aminoethyltetrazole **4a-L** and the d-enantiomer **4a-D** displayed no activity when tested at concentrations of up to 128 mg/L against our panel of clinically relevant bacteria. As already stated, the natural amino acid alanine is transported as oligopeptide forms into bacterial cells, and not as a single amino acid [20]. This latest result suggests that the increase in lipophilicity associated with changing the carboxylate in the natural compound to the tetrazole in our analogue does not produce effective passive diffusion at an extent that would lead to significant antimicrobial activity. Furthermore, l-1-aminopropyltetrazole **4b-L** was also inactive when exposed to the bacteria as the single amino acid analogue.

In marked contrast to the results with the single amino acid analogues, targeting the dipeptide permease systems of bacteria proved to be successful and is consistent with the intracellular liberation of l-1-aminoethyltetrazole **4a**-L (Table 2). Depending upon the N-terminal amino acid, the growth of different Gram-negative and Gram-positive bacteria can be inhibited, thus selectivity can be tuned by choosing specifiC-Natural N-terminal amino acids within the oligopeptide analogues.

Importantly, it was shown that l,l-alanyl peptide **14a-LL** was effective in inhibiting the growth of many Gram-negative species, but did not display inhibition against *Salmonella*. This pattern of activity leads to an obvious and important application for this compound; it could be used in *Salmonella* selective media to improve the clinical detection of *Salmonella* from stool samples, in which overgrowth by commensal gut bacteria, such as *E. coli*, is problematic [1].

The glycyl peptide **14g-L** was only active against a few species, and even then, only in higher concentrations. In addition, the β-alanyl **14d-L** substrate did not show any antibacterial activity. This compound was aimed at targeting β-alanyl aminopeptidase, which only presents in a few bacteria, most notably *Pseudomonas aeruginosa* [21], *Serratia marcescens* and *Burkholderia cepacia* [22]. This result suggests that either the side chain or the chirality of the N-terminal amino acid might have an important role in their transportation. Previously reported corresponding β-alanyl derivatives of the AlaR inhibitor fosfalin and β-chloroalanine also had limited activities against bacteria, despite other peptide derivatives of these compounds being well known effective and selective antibacterial agents [23,24,25].

The compounds displaying longer, more lipophilic side chains i.e., the peptide analogues l,l-norvalyl **14e-LL**, leucyl **14h-LL**, isoleucyl **14i-LL** and phenylalanyl **14j-LL** displayed enhanced antibacterial activity against Gram-positive bacteria, and suggest roles as new wide spectrum and selective antibacterial agents. However, what the current data set does not tell us is why this enhancement in activity is seen.

One possibility is that activity against Gram-positive bacteria may occur due to the further lipophilicity associated with the side chains of this group of substrates. This increased lipophilicity could promote passive diffusion as well as, or instead of, active transport into bacteria. Alternatively, the enhanced activity of the peptides displaying longer side chains is consistent with the reported increased affinity for large lipophilic side chains with bacterial proton-dependent oligopeptide transporter proteins [26]. As l,l-leucyl **14h-LL**, isoleucyl **14i-LL** and phenylalanyl **14j-LL** peptides selectively allow the growth of *Salmonella* species in low concentrations, they also have potential use in a new improved *Salmonella* selective medium. Another useful finding present within the data is the selective inhibition of enterococci by l-pyroglutamyl derivative **14f-LL** [27]. This pyroglutamyl derivative was selected and synthesized to target enterococci, as they are known to produce an active pyroglutamyl aminopeptidase enzyme [28]. The great selectivity displayed by this compound could be attributed to the amide motif on the N-terminus, as the N-terminal amino group is necessary for the transport carried out by permease systems [20]. Interestingly, the d-1-aminoethyltetrazole containing peptides **14a-LD**, **14h-LD** were totally inactive, although their l-1-aminoethyltetrazole containing diastereomers **14a-LL**, **14h-LL** inhibited many different species (Table 3). This again suggests an active transport mode of entry wherein the transporter would potentially display chiral selectivity.

l-Alanyl-l-1-aminopropyltetrazole **14b-LL** was found to only inhibit *Enterobacter cloacae*, among the panel of bacteria that were tested. The contrast between the results obtained with l-alanyl-l-1-aminopropyltetrazole **14b-LL** and l-alanyl-l-1-aminoethyltetrazole **14a-LL** represents the difference between l-1-aminoethyltetrazole **4a-L** and l-1-aminopropyltetrazole **4b-L** homologues, as l-1-aminoethyltetrazole **4a-L** displays a bioisosteric replacement of the natural substrate of AlaR and can inhibit the enzyme in different species. The larger homologue **4b-L** probably has limited access to the active site of the enzyme in most bacterial AlaRThe tripeptide analogue l-Alanyl-l-alanyl-l-1-aminoethyltetrazole **14c-LLL** displayed a similar profile to the previously discussed dipeptides **14a-LL** and **14h-LL**, which also suggests potential in media where *Salmonella* detection is required. The N-terminal succinyl containing peptides **16-LL**, **17-LLL** and **18-LLLL** did not produce significant inhibitory effects against most bacteria, Table 3.

## 3. Materials and Methods

### 3.1. General Information

All commercially available reagents and solvents were obtained from Sigma-Aldrich Dorset, UK), Apollo Scientific (Stockport, UK), Alfa Aesar (Heysham, UK), Fluorochem (Glossop, UK), or Fischer Scientific (Loughborough, UK) and were used without further purification. Melting points were recorded on a Reichart-Kofler hot-stage microscope apparatus (Reichart, Vienna, Austria) and are uncorrected. Infrared spectra were recorded in the range 4000–600 cm^−1^ using a Spectrum BX FT-IR spectrophotometer (Perkin Elmer, Beaconsfield, UK). NMR spectra were obtained using an Ultrashield 300 spectrometer (Bruker, Coventry, UK) at 300 MHz for ^1^H spectra or at 75 MHz for ^13^C spectra and a Bruker Avance III Ultrashield spectrometer at 500 MHz for ^1^H spectra or at 125 MHz for ^13^C spectra. Low-resolution mass spectra were recorded on a Bruker Esquire 3000plus analyser using an electrospray source in either positive or negative ion mode. High resolution accurate mass measurements were collected by the EPSRC-National Mass Spectrometry Facility at Swansea University. Elemental analyses were performed using an CE-440 Elemental Analyzer (Exeter Analytical, Coventry, UK). Thin layer chromatography was performed on silica gel 60F254 (Merck, Hoddesdon, UK). Thin layer chromatography results were analysed by UV lamp (254 nm), ninhydrin stain (amine, amide content) and/or ceric ammonium molybdate stain (oxidising agent). Fischer silica gel 60 (35-70 micron, Fischer Scientific, Loughborough, UK) was used for column chromatography; the samples were pre-absorbed onto silica 60 (35-70 micron). LC-MS analysis was performed using a 1290 Infinity Series HPLC system (Agilent, Waldbronn, Germany) and an Agilent 6120 Quadrupole LC-MS detector. LC-MS data was analysed by an Agilent ChemStation. Where required, compounds were purified by Agilent 1260 preparative HPLC. Hydrogenation reactions were performed in a Parr 4560 mini benchtop reactor (SciMed, Stockport, UK). Air and moisture sensitive reactions were carried out in oven dried glassware under a nitrogen atmosphere. 

### 3.2. Chemistry

#### 3.2.1. General Method A: Synthesis of Protected Amino Acid Methyl Ester:

To a solution of N-protected-amino acid (26.4 mmol) in dry DMF (70 mL) caesium carbonate (4.72 g, 14.5 mmol) was added and the mixture was stirred for 30 min at room temperature. After dropwise addition of methyl iodide (1.73 mL, 27.8 mmol), the reaction was stirred overnight. Ethyl acetate (50 mL) was then added to the mixture and it was washed with water (3 × 50 mL), then the combined aqueous phase was extracted with ethyl acetate (50 mL). The combined organic phase was washed with 10% K_2_CO_3_ solution (2 × 30 mL), then with brine (30 mL), dried over MgSO_4_, and subsequently filtered. The solvent was removed under reduced pressure to give the product as a white solid.

#### 3.2.2. General Method B: Synthesis of Protected Amino Amides:

The protected amino acid methyl ester (420 mmol) was dissolved and stirred in 7 M solution of ammonia in methanol (60 mL) overnight at 50 °C in a sealed tube. After the reaction was complete, as determined by TLC, the mixture was evaporated to dryness under reduced pressure.

#### 3.2.3. General Method C: Synthesis of Protected Amino Nitrile:

To a solution of the required N-protected-alanineamide (19.0 mmol) in DCM (9 mL), pyridine (14 mL) was added. After cooling to 0 °C, tosyl chloride (4.7 g, 24.7 mmol) was added to the solution. Stirring was continued for 30 min at 0 °C and then overnight at room temperature. After the reaction as complete, as determined by TLC, the system was cooled to 0 °C and quenched with water (70 mL) nd ethyl acetate (70 mL). After separation, the aqueous layer was extracted with ethyl acetate (4 × 30 mL) and the combined organic phase was washed with 1.2 M HCl (3 × 40 mL), and saturated NaHCO_3_ (50 mL), then brine (50 mL). The whole solution was dried over MgSO_4_, and evaporated under reduced pressure.

#### 3.2.4. General Method D: Synthesis of Cbz-1-Aminoalkyltetrazole:

To a suspension of NH_4_Cl (2.03 g, 38 mmol) and NaN_3_ (2.62 g, 38 mmol) in dry DMF (60 mL) in a two-necked round bottom flask equipped with reflux condenser and drying tube, the required Cbz-amino nitrile (38 mmol) was added. The mixture was stirred and heated to 90 °C for 1 h. To the cooled mixture, another portion of NH_4_Cl (1.04 g, 19 mmol) and NaN_3_ (1.31 g, 19 mmol) were added and the mixture was heated at 90 °C overnight. When the reaction was complete, as determined by TLC, the mixture was filtered and the residue was washed with ethyl acetate. The filtrate was evaporated under reduced pressure. Water (180 mL) was added to the residue and it was acidified to pH 1 with 2.5 M HCl aqueous solution. The precipitated solid was filtered off and washed with water to give the product.

#### 3.2.5. General Method E: Removal of the Cbz Protecting Group:

To the solution of the protected amino acid or peptide derivative (1.0 meq) in methanol, 5% palladium on activated carbon (0.1 meq) was added portion wise. The mixture was stirred at room temperature under 2 bar pressure of hydrogen overnight in an autoclave. After the reaction was complete determined by TLC, the mixture was filtered through celite and then the solvent removed in vacuum.

#### 3.2.6. General Method F: Fmoc Protection:

10% Aqueous Na_2_CO_3_ (13 mL) was added to the suspension of 1-aminolakyltetrazole (4.87 mmol) in 1,4-dioxane (7 mL). After the addition of Fmoc chloride (1.38 g, 5.35 mmol) solution in 1,4-dioxane (7 mL), the mixture was stirred for 30 min at 0 °C, then overnight at room temperature. When the reaction was complete, as determined by TLC, it was acidified with 2.5 M HCl aqueous solution (11 mL). The mixture was filtered and the solid product was washed with water before drying.

#### 3.2.7. General Method G: Solid Phase Peptide Coupling:

Swelling: The resin was bubbled for 30 min in a long-necked glass sinter in a solvent (DMF or DCM) with a volume 3x that of the bed volume. Subsequently the solvent was removed by filtration.

Washing off: The resin was washed with DMF (3 × 8 mL for 30 s), IPA (2 × 8 mL for 30 s) and finally with petroleum ether (40–60 °C) (2 × 8 mL for 30 s). The resin was air-dried, then dried in a vacuum oven at 40 °C overnight. The solvents were removed by filtration after each step.

Attachment to 2-chlorotrityl chloride resin: After swelling the resin in DCM the solution of Fmoc-1-aminoalkyltetrazole 4 and DIPEA in DCM was added to the resin, and the mixture was bubbled for 2 h. After filtration, the resin was bubbled with DCM, DIPEA and methanol mixture for 1 h. Later the resin was washed with DMF (3 × 8 mL for 30 s), methanol (3 × 8 mL for 30 s) then with DCM or petroleum ether (40–60 °C) (3 × 8 mL for 30 s) depending on whether the resin needed to be swelled or dried. The solvents and solutions were removed by filtration after each step.

Fmoc deprotection: The resin swelled in DMF was treated with a mixture of piperidine and DMF (1:4 ratio, 10 mL/g resin, 1 × 5 min and 3 × 10 min), then it was alternately washed with DMF and IPA (2 × 8 mL for 30 s) finished with washing with DMF (8 mL for 30 s). The solvents and solutions were removed by filtration after each step.

Fmoc test: A sample consisting of a few resin beads was washed with DMF (4 × 1 mL for 30 s), IPA (5 × 1 mL for 30 s) and with methanol (2 × 1 mL for 30 s), followed by transfer into a separate flask by washing with 1% TFA in DCM (2 × 1 mL for 30 s). The filtrate was placed on a TLC plate with a capillary and observed under UV light (254 nm). A positive test shows fluorescence on the TLC plate. The solvents and solutions were removed by filtration after each step.

Coupling: DIPEA (1 eq) was added to the solution of HBTU (3 eq) and the protected amino acid (3 eq) in DMF (10 mL). The solution was stirred for 8 min and then it was added to the swelled resins. After 20 min bubbling witH-Nitrogen, more DIPEA (0.5 eq) was added to the mixture, then it was bubbled until the TNBS test showed a negative result. After the reaction was complete the resin was washed off using the procedure mentioned above.

Coupling with succinic anhydride: The solution of succinic anhydride (3 eq) in DMF (10 mL) was added to the swelled resins, followed by the addition of DIPEA (1 eq). It was bubbled until the TNBS test showed a negative result (typically 1 h). After the reaction was over the resin was washed off using the procedure mentioned above.

TNBS test: The solution of DIPEA in DMF (10%, 3 drops) and the aqueous solution of TNBS (1 M, 3 drops) were mixed with sample consisting of a few resin beads. After a waiting period of 10 min, the resin beads were inspected under a light microscope (magnification 4×). The result was positive if the beads were coloured red, orange or yellow and it was negative if the beads were colourless.

Cleavage from resin: The resin swelled in DCM was treated with 1% TIS containing TFA, DMF mixture (5:95 ratio, 10 mL/g) for 1 h. After washing the resin with the same TFA, DCM mixture, the filtrate was evaporated under reduced pressure. (Appendix A).

HPLC method A: column: ACE 5 C18 (150 × 3 mm i.d.); eluent: isocratic: water (0.1% formic acid) - methanol (0.1% formic acid) 95:5 ratio. The purity of all the compounds tested using this method were ≥95%.

HPLC method B: column: ACE 5 C18 (150 × 3 mm i.d.); eluent: isocratic: water (0.1% formic acid) - methanol (0.1% formic acid) 80:20 ratio. The purity of all the compounds tested using this method were ≥95%.

HPLC method C: column: ACE 5 C18 (150 × 3 mm i.d.); eluent: isocratic: water (0.1% formic acid) - methanol (0.1% formic acid) 50:50 ratio. The purity of all the compounds tested using this method were ≥95%.

HPLC method D: column: ACE 5 C18 (150 × 3 mm i.d.); eluent: isocratic: water (0.1% formic acid) - methanol (0.1% formic acid) 20:80 ratio. The purity of all the compounds tested using this method were ≥95%.

Prep HPLC method A: column: Vydac protein and peptide 218TP510; eluent: gradient: water (0.1% formic acid) - methanol (0.1% formic acid) (0–90%).

*Methyl-(2S or R)-2-{[(benzyloxy)carbonyl]amino}propanoate* (**6a**) [29]: Cbz-l/d-alanine **5a** (10.0 g, 45.0 mmol) was esterified and worked up using general method A, yielding the desired product as a white powder (10.6 g, 99%). The reaction and the work up were monitored by TLC (eluent: petroleum ether (40–60 °C) - ethyl acetate, 2:1 ratio, R_f_: 0.51). Mp: 41–43 °C (literature: 43–44 °C, light petrol)[30]; ^1^H-NMR (300 MHz, DMSO-*d*_6_) δ_H_ 7.74 (1H, d, *J* = 7.5 Hz, 5-H), 7.34 (5H, m, 9,10,11-H), 5.03 (2H, s, 7-H), 4.11 (1H, quint, *J* = 7.5 Hz, 2-H), 3.63 (3H, s, 4-H), 1.27 (3H, d, *J* = 7.2 Hz, 3-H); ^13^C-NMR (75.5 MHz, DMSO-*d*_6_) δ_C_ 173.8 (1-C), 156.3 (6-C), 137.4 (8-C), 128.8 (2C, 10-C), 128.3 (11-C), 128.2 (2C, 9-C), 65.9 (7-C), 52.3 (4-C), 49.8 (2-C), 17.4 (3-C); ν_max_/cm^−1^ 3335 (N-H), 1703 (C=O), 1668 (C=O), 1524 (N-H), 1209 (C-O); MS(ESI) *m*/*z* 260.1 (M + Na)^+^.

*Benzyl-N-[(1S or R)-1-carbamoylethyl]carbamate* (**7a**) **[9]**: Cbz-l/d-alanine methyl ester **6a** (10.0 g, 42 mmol) was reacted and worked up using general method B, to give the product as a white powder (9.40 g, 99%). The reaction and the work up were monitored by TLC (eluent: petroleum ether (40–60 °C) - ethyl acetate, 1:1 ratio, R_f_: 0.11). Mp: 131–132 °C (literature: 130–131 °C, methanol–diethyl ether)[31]; ^1^H-NMR (300 MHz, DMSO-*d*_6_) δ_H_ 7.35 (6H, m, 5,9,10,11-H), 7.26 (1H, s, 4-H_a_), 6.93 (1H, s, 4-H_b_), 5.02 (2H, s, 7-H), 3.98 (1H, quint, *J* = 7.2 Hz, 2-H), 1.21 (3H, d, *J* = 7.2 Hz, 3-H); ^13^C-NMR (75.5 MHz, DMSO-*d*_6_) δ_C_ 174.9 (1-C), 156.1 (6-C), 137.5 (8-C), 128.76 (2C, 10-C), 128.2 (11-C), 128.1 (2C, 9-C), 65.8 (7-C), 50.4 (2-C), 18.7 (3-C); ν_max_/cm^−1^ 3386 (N-H), 3308 (N-H), 3196 (N-H), 1651 (C=O), 1537 (N-H), 1247 (C-O); MS(ESI) *m*/*z* 223.1 (M + H)^+^.

*Benzyl-N-[(1S or R)-1-cyanoethyl]carbamate* (**8a**) [9]**:** Cbz-l/d-alanine amide **7a** (3.48 g, 15.7 mmol) was reacted and worked up using general method C. The crude product was washed with heptane to give the product as white crystals (3.12 g, 97%). The reaction and the work up were monitored by TLC (eluent: petroleum ether (40–60 °C) - ethyl acetate 1:1 ratio, R_f_: 0.41). Mp: 79–80 °C (literature: 82–83 °C [32]); ^1^H-NMR (300 MHz, DMSO-*d*_6_) δ_H_ 8.14 (1H, d, *J* = 5.4 Hz, 4-H), 7.37 (5H, m, 8,9,10-H), 5.08 (2H, s, 6-H), 4.59 (1H, quint, *J* = 7.2 Hz, 2-H), 1.43 (3H, d, *J* = 7.2 Hz, 3-H); ^13^C-NMR (75.5 MHz, DMSO-*d*_6_) δ_C_ 155.7 (5-C), 137.0 (7-C), 128.9 (9-C), 128.5 (10-C), 128.4 (8-C), 120.9 (1-C), 66.5 (6-C), 38.2 (2-C), 18.8 (3-C); ν_max_/cm^−1^ 3332 (N-H), 2356 (C≡N), 1686 (C=O), 1521 (N-H), 1253 (C-O); MS(ESI) *m*/*z* 227.1 (M + Na)^+^.

*Benzyl-N-[(1S or R)-1-(1H-1,2,3,4-tetrazol-5-yl)ethyl]carbamate* (**9a**) **[9]**: Cbz-l/d-alanyl nitrile (**8a**, 7.80 g, 38.0 mmol) was reacted and worked up using general method D. The product was gained as a white solid (7.55 g, 80%). The reaction and the work up were monitored by TLC (eluent: petroleum ether (40–60 °C) - ethyl acetate 1:1, R_f_: 0.09). Mp: 134–138 °C (literature: 139–141 °C [33]); ^1^H-NMR (300 MHz, DMSO-*d*_6_) δ_H_ 7.99 (1H, d, *J* = 6.9 Hz, 4-H), 7.33 (5H, m, 9,10,11-H), 5.01 (3H, m, 2,7-H), 1.49 (3H, d, *J* = 7,2 Hz, 3-H); ^13^C-NMR (75.5 MHz, DMSO-*d*_6_) δ_C_ 158.6 (1-C), 155.7 (6-C), 136.8 (8-C), 128.31 (2C, 10-C), 127.8 (3C, 9,11-C), 65.7 (7-C), 41.9 (2-C), 19.3 (3-C); ν_max_/cm^–1^ 3308 (N-H), 2980, 2866, 2738, 2614 (tetrazole), 1686 (C=O), 1524 (N-H), 1255 (C-O); MS(ESI) *m*/*z* 248.1 (M + H)^+^.

*(1S or R)-1-(1H-1,2,3,4-tetrazol-5-yl)ethan-1-amine* (**4a**) [9]: Cbz-l/d-aminoethyltetrazole (**9a**, 0.75 g, 3.0 mmol) was deprotected using general method E. Following filtration, the solid residue was washed with ethanol and the filtrate was evaporated under reduced pressure to give the product as a white solid (0.33 g, 97%). The reaction and the work up were monitored by TLC (eluent: ethyl acetate - ethanol 8:1 ratio, R_f_: 0.03). Mp: 230 °C (decomposed) (literature: 268–270 °C, (decomposed) **[9]**). ^1^H-NMR (300 MHz, DMSO-*d*_6_) δ_H_ 4.55 (1H, quart, *J* = 6.6 Hz, 2-H), 1.54 (3H, d, *J* = 6.9 Hz, 3-H); ^13^C-NMR (75.5 MHz, DMSO-*d*_6_) δ_C_ 160.4 (1-C), 44.1 (2-C), 19.8 (3-C); ν_max_/cm^–1^ 3388 (N-H), 2916, 2719, 2634, 2522 (tetrazole); MS(ESI) *m*/*z* 114.1 (M + H)^+^.

*Methyl-(2S)-2-{[(benzyloxy)carbonyl]amino}butanoate* (**6b-L**) [34]: Cbz-l-ethylglycine (**5b-L**, 5.0 g, 21.0 mmol) was reacted and worked up using general method A, yielding the product as colourless oil (5.1 g, 97%). The reaction and the work up were monitored by TLC (eluent: petroleum ether (40–60 °C) - ethyl acetate, 2:1 ratio, R_f_: 0.77). ^1^H-NMR (300 MHz, DMSO-*d*_6_) δ_H_ 7.69 (1H, d, *J* = 7.5 Hz, 6-H), 7.36 (5H, m, 10,11,12-H), 5.04 (2H, s, 8-H), 3.97 (1H, td, *J* = 5.4, *J* = 8.4 Hz, 2-H), 3.63 (1H, s, 5-H), 1.66 (2H, m, 3-H), 0.89 (3H, t, *J* = 7.2 Hz, 4-H); ^13^C-NMR (75.5 MHz, DMSO-*d*_6_) δ_C_ 173.2 (1-C), 156.6 (7-C), 137.4 (9-C), 128.8 (11-C), 128.3 (12-C), 128.2 (10-C), 65.9 (8-C), 55.8 (2-C), 52.2 (5-C), 24.6 (3-C), 10.9 (4-C); ν_max_/cm^–1^ 3345 (N-H), 1703 (C=O), 1520 (N-H), 1206 (C-O); MS(ESI) *m*/*z* 250.8 (M – H)^-^.

*Benzyl-N-[(1S)-1-carbamoylpropyl]carbamate* (**7b-L**) [35]: Cbz-l-ethylglycine methyl ester (**6b-L**, 2.0 g, 8.0 mmol) was reacted and worked up using general method B. The crude product was washed with diethyl ether to give the product as white crystals (1.72 g, 91%). The reaction and the work up were monitored by TLC (eluent: petroleum ether (40–60 °C) - ethyl acetate, 1:1 ratio, R_f_: 0.07). Mp: 140–142 °C (literature: 141 °C [36]); ^1^H-NMR (300 MHz, DMSO-*d*_6_) δ_H_ 7.36 (5H, m, 10,11,12-H), 7.29 (1H, s, 5a-H), 7.20 (1H, d, *J* = 8.4 Hz, 6-H), 6.95 (1H, s, 5b-H), 5.03 (2H, s, 8-H), 3.87 (1H, td, *J* = 5.4, *J* = 8.4 Hz, 2H), 1.67 (1H, m, 3a-H), 1.55 (1H, m, 3b-H), 0.86 (3H, t, *J* = 7.2 Hz, 4-H); ^13^C-NMR (75.5 MHz, DMSO-*d*_6_) δ_C_ 174. 2 (1-C), 156.4 (7-C), 137.6 (9-C), 128.8 (11-C), 128.2 (12-C), 128.1 (10-C), 65.8 (8-C), 56.3 (2-C), 25.6 (3-C), 10.8 (4-C); ν_max_/cm^–1^ 3380 (N-H), 3308 (N-H), 3196 (N-H), 1649 (C=O), 1535 (N-H), 1240 (C-O); MS(ESI) *m*/*z* 237.1 (M + H)^+^.

*Benzyl-N-[(1S)-1-cyanopropyl]carbamate* (**8b-L**) [37]: Cbz-l-ethylglycine amide (**7b-L**, 4.25 g, 18.0 mmol) was reacted and worked up using general method C to give the product as white crystals (3.88 g, 99%). The reaction and the work up were monitored by TLC (eluent: petroleum ether (40–60 °C) - ethyl acetate 1:2 ratio, R_f_: 0.64). Mp: 43–45 °C; ^1^H-NMR (300 MHz, DMSO-*d*_6_) δ_H_ 8.15 (1H, d, *J* = 7.8 Hz, 5-H), 7.37 (5H, m, 9,10,11-H), 5.09 (2H, s, 7-H), 4.48 (1H, quart, *J* = 7.8 Hz, 2-H), 1.78 (2H, pentd, *J* = 1.5, *J* = 7.5 Hz, 3-H), 0.96 (3H, t, *J* = 7.2 Hz, 4-H); ^13^C-NMR (75.5 MHz, DMSO-*d*_6_) δ_C_ 156.0 (6-C), 137.0 (8-C), 128.9 (10-C), 128.5 (11-C), 128.5 (9-C), 120.1 (1-C), 66.6 (7-C), 44.2 (2-C), 25.8 (3-C), 10.3 (4-C); ν_max_/cm^–1^ 3312 (N-H), 1694 (C=O), 1524 (N-H), 1263 (C-O); MS(ESI) *m*/*z* 219.1 (M + H)^+^.

*Benzyl-N-[(1S)-1-(1H-1,2,3,4-tetrazol-5-yl)propyl]carbamate* (**9b-L**) [38]: Cbz-l-ethylglycyl nitrile (**8b-L**, 3.9 g, 18 mmol) was reacted and worked up using general method D. The product was gained as a white solid (3.83 g, 81%). The reaction and the work up were monitored by TLC (eluent: petroleum ether (40–60 °C) - ethyl acetate 1:2, R_f_: 0.20). Mp: 142–144 °C ; ^1^H-NMR (300 MHz, DMSO-*d*_6_) δ_H_ 7.98 (1H, d, *J* = 7.2 Hz, 6-H), 7.36 (5H, m, 10,11,12-H), 5.08 (1H, d, *J* = 12.3 Hz, 8a-H), 5.02 (1H, d, *J* = 12.6 Hz, 8b-H), 4.83 (1H, quart, *J* = 8.1 Hz, 2-H), 1.89 (2H, m, 3-H), 0.89 (3H, t, *J* = 7.2 Hz, 4-H); ^13^C-NMR (75.5 MHz, DMSO-*d*_6_) δ_C_ 158.4 (1-C), 156.4 (7-C), 137.3 (9-C), 128.8 (11-C), 128.3 (12-C), 128.2 (10-C), 66.1 (8-C), 48.2 (2-C), 26.7 (3-C), 10.7 (4-C); ν_max_/cm^–1^ 3296 (N-H), 2976, 2880, 2714, 2615, (tetrazole), 1686 (C=O), 1528 (N-H), 1263 (C-O); MS(ESI) *m*/*z* 262.1 (M + H)^+^.

*(1S)-1-(1H-1,2,3,4-tetrazol-5-yl)propane-1-amine* (**4b-L**) [39]: Cbz-l-aminopropyltetrazole (**9b-L**, 3.72 g, 14.0 mmol) was deprotected using general method E. Following filtration, the solid residue was washed with ethanol and the filtrate was evaporated under reduced pressure to give the crude, which was washed to gain the pure product as a white solid (1.60 g, 88%). The reaction and the work up were monitored by TLC (eluent: ethanol, R_f_: 0.18). Mp: 260 °C, decomposed (literature: 273–274 °C [39]). ^1^H-NMR (300 MHz, D_2_O) δ_H_ 4.62 (1H, t, *J* = 7.2 Hz, 2-H), 2.05 (2H, quint, *J* = 7.5 Hz, 3-H), 0.82 (3H, t, *J* = 7.5 Hz, 4-H); ^13^C-NMR (75.5 MHz, D_2_O) δ_C_ 159.6 (1-C), 48.7 (2-C), 26.0 (3-C), 8.9 (4-C); ν_max_/cm^–1^ 2820, 2716, 2613, 2561 (tetrazole), 1530 (N-H); MS(ESI) *m*/*z* 128.1 (M + H)^+^.

*9H-Fluoren-9-ylmethyl-N-[(1S or R)-1-(1H-1,2,3,4-tetrazol-5-yl)ethyl]carbamate* (**10a**) [40]: l/d-1-Aminoethyltetrazole (**4a**, 2.00 g, 17.7 mmol) was reacted using general method F. The crude product was purified by recrystallization from acetonitrile to give the pure product as white crystals (0.92 g, 56%). The reaction and the work up was monitored by TLC (eluent: dichloromethane – ethanol 20:1 ratio, R_f_: 0.07). Mp: 197–199 °C (literature: 200–202 °C (decomposed) [40]); ^1^H-NMR (300 MHz, DMSO-*d*_6_) δ_H_ 8.05 (1H, d, *J* = 7.5 Hz, 4-H), 7.88 (2H, d, *J* = 7.5 Hz, 13-H), 7.71 (2H, t, *J* = 6.6 Hz, 10-H), 7.41 (2H, t, *J* = 7.2 Hz, 12-C), 7.31 (2H, m, 11-H), 5.01 (1H, quint, *J* = 7.2 Hz, 2-H), 4.32 (1H, m, 8-H), 4.24 (2H, m, 7-H), 1.51 (3H, d, *J* = 6.9 Hz, 3-H); ^13^C-NMR (75.5 MHz, DMSO-*d*_6_) δ_C_ 159.0 (1-C), 156.0 (6-C), 144.3 (9a-C), 144.2 (9b-C), 141.2 (14-C), 128.1 (12-C), 127.5 (11-C), 125.7 (10-C), 120.6 (13-C), 66.2 (7-C), 47.1 (8-C), 42.3 (2-C), 19.8 (3-C); ν_max_/cm^–1^ 3310 (N-H), 2965, 2868, 2740, 2617 (tetrazole), 1686 (C=O), 1523 (N-H), 1258 (C-O); MS(ESI) *m*/*z* 336.2 (M + H)^+^.

*9H-Fluoren-9-ylmethyl-N-[(1S)-1-(1H-1,2,3,4-tetrazol-5-yl)propyl]carbamate* (**10b-L**): l-1-Amino-ethyltetrazole (**4b-L**, 1.79 g, 6.9 mmol) was reacted using general method F. The crude product was purified by recrystallization in 2 generations from acetonitrile to give the pure product as white crystals (1.66 g, 75%). The reaction and the work up were monitored by TLC (eluent: dichloromethane – ethanol 10:1 ratio, R_f_: 0.21). Mp: 209–211 °C (melted and decomposed); ^1^H-NMR (300 MHz, DMSO-*d*_6_) δ_H_ 16.26 (6-H), 8.04 (1H, d, *J* = 7.8 Hz, 5-H), 7.89 (2H, d, *J* = 7.5 Hz, 14-H), 7.72 (2H, t, *J* = 7.2 Hz, 11-H), 7.42 (2H, t, *J* = 7.5 Hz, 13-H), 7.33 (2H, m, 12-H), 4.81 (1H, quart, *J* = 8.1 Hz, 2-H), 4.34 (2H, m, 8-H), 4.23 (1H, m, 9-H), 1.91 (2H, m, 3-H), 0.88 (3H, t, *J* = 7.2 Hz, 4-H); ^13^C-NMR (75.5 MHz, DMSO-*d*_6_) δ_C_ 156.4 (7-C), 144.3 (10a-C), 144.2 (10b-C), 141.2 (15-C), 128.1 (13-C), 127.5 (12-C), 125.7 (11-C), 120.6 (14-C), 66.2 (8-C), 48.1 (2-C), 47.1 (9-C), 26.6 (3-C), 10.7 (4-C); ν_max_/cm^–1^ 3300 (N-H), 2978, 2876, 2717, 2617 (tetrazole), 1682 (C=O), 1533 (N-H), 1265 (C-O); MS(ESI) *m*/*z* 350.1 (M + H)^+^; CHN [Found: C, 65.08; H, 5.53; N, 19.86. C_19_H_19_N_5_O_2_ requires C, 65.32; H, 5.48; N, 19.86%].

*Benzyl-N-({[(1S)-1-(1H-1,2,3,4-tetrazol-5-yl)ethyl]carbamoyl}methyl)carbamate* (**13g-L**): Prepared using general method G. The Fmoc group of Fmoc-1-l-aminoethyltetrazole attached to 2-chlorotrityl chloride resin (resin:loading; 1:1 ratio; 1.3 mmol) was removed, this was monitored by the Fmoc and TNBS tests. After the coupling of Cbz-glycine with the resin, the title compound was cleaved off. The cleaving mixture (TFA/DCM/TIS, ratio 5:95:1 eq) was removed by evaporation under reduced pressure. The residue washed with petroleum ether (40 – 60 °C) to obtain the product as white crystals (0.24 g, 59%). The reaction and the work up were monitored by TLC (eluent: ethanol, R_f_: 0.56). Mp: 134–136 °C; ^1^H-NMR (300 MHz, DMSO-*d*_6_) δ_H_ 8.56 (1H, d, *J* = 7.2 Hz, 4-H), 7.35 (6H, m, 7,12,13,14-H), 5.24 (1H, quint, *J* = 7.2 Hz, 2-H), 5.03 (2H, s, 10-H), 3.68 (2H, d, *J* = 6.3 Hz, 6-H), 1.49 (3H, d, *J* = 6.9 Hz, 3-H); ^13^C-NMR (75.5 MHz, DMSO-*d*_6_) δ_C_ 169.3 (5-C), 158.9 (1-C), 156.9 (9-C), 137.5 (11-C), 128.8 (13-C), 128.2 (14-C), 128.1 (12-C), 65.9 (10-C), 43.8 (6-C), 40.3 (2-C), 19.8 (3-C); ν_max_/cm^–1^ 3289 (N-H), 3269 (N-H), 1697 (C=O), 1661 (C=O), 1530 (N-H), 1244 (C-O); MS(ESI) *m*/*z* 305.1 (M + H)^+^; HRMS (Found (M + H)^+^ 305.1360. Calcd. for C_13_H_17_O_3_N_6_: (M + H)^+^ 305.1357.); (Purity test: HPLC method D).

2*-Amino-N-[(1S)-1-(1H-1,2,3,4-tetrazol-5-yl)ethyl]acetamide* (**14g-L**): Cbz-glycyl-1-l-aminoethyl-tetrazole (**13g-L**, 0.18 g, 0.57 mmol) was deprotected by general method E. Following filtration, the solid residue was washed with 30 mL methanol, which was discarded. The solid residue was washed into a separate flask with water, which was freeze dried to give the product as a white solid (0.01 g, 99%). Further purification was performed by Prep HPLC method A. Mp: 180 °C (decomposed); ^1^H-NMR (300 MHz, D_2_O) δ_H_ 5.27 (1H, quart, *J* = 6.9 Hz, 2-H), 3.75 (2H, s, 6-H), 1.52 (3H, d, *J* = 6.9 Hz, 3-H); ^13^C-NMR (75.5 MHz, D_2_O) δ_C_ 167.3 (5-C), 164.3 (1-C), 42.3 (2-C), 41.0 (6-C), 19.3 (3-C); ν_max_/cm^–1^ 3215 (N-H), 1672 (C=O) 1547 (N-H); MS(ESI) *m*/*z* 171.1 (M + H)^+^; HRMS (Found (M + H)^+^ 171.0987. Calcd. for C_5_H_11_ON_6_: (M + H)^+^ 171.0989.); (Purity test: HPLC method A).

*Benzyl-N-[(1S)-1-{[(1S)-1-(1H-1,2,3,4-tetrazol-5-yl)ethyl]carbamoyl}ethyl]carbamate* (**13a-LL**): Prepared using general method G. The Fmoc group of Fmoc-1-l-aminoethyltetrazole attached to 2-chlorotrityl chloride resin (resin:loading; 1:1 ratio; 1.3 mmol) was removed, this was monitored by the Fmoc and TNBS tests. After the coupling of Cbz-l-alanine with the resin, the title compound was cleaved off. The cleaving mixture (TFA/DCM/TIS, ratio 5:95:1 eq) was removed by evaporation under reduced pressure and washed with petroleum ether (40–60 °C) to obtain the product as white crystals (0.23 g, 58%). The reaction and the work up were monitored by TLC (eluent: ethyl acetate and 1 drop of glacial acetic acid, R_f_: 0.07). Mp: 186–187 °C; ^1^H-NMR (300 MHz, DMSO-d_6_) δ_H_ 16.18 (1H, br s, 9-H), 8.54 (1H, d, *J* = 7.5 Hz, 4-H), 7.35 (6H, m, 8,13-15-H), 5.23 (1H, quint, *J* = 7.2 Hz, 2-H), 5.05 (1H, d, *J* = 12.6 Hz, 11-H_a_), 5.00 (1H, d, *J* = 12.6 Hz, 11-H_b_), 4.09 (1H, quint, *J* = 7.2 Hz, 6-H), 1.50 (3H, d, *J* = 7.2 Hz, 3-H), 1.20 (3H, d, *J* = 7.2 Hz, 7-H); ^13^C-NMR (75.5 MHz, DMSO-d_6_) δ_C_ 172.1 (5-C), 158.3 (1-C), 155.6 (10-C), 137.0 (12-C), 128.3 (14-C), 127.7 (15-C), 127.7 (13-C), 65.3 (11-C), 49.8 (6-C), 40.3 (2-C), 19.2 (7-C), 17.9 (3-C); ν_max_/cm^–1^ 3287 (N-H), 3264 (N-H), 1684 (C=O), 1655 (C=O), 1535 (N-H), 1230 (C-O); MS(ESI) *m*/*z* 319.2 (M + H)^+^; CHN [Found: C, 53.06; H, 5.64; N, 26.58. C_14_H_18_N_6_O_3_ requires C, 52.82; H, 5.70; N, 26.40%].

*(2S)-2-amino-N-[(1S)-1-(1H-1,2,3,4-tetrazol-5-yl)ethyl]propanamide* (**14a-LL) [19]**: Cbz-l-alanyl-1-l-aminoethyltetrazole (**13a-LL**, 0.58 g, 1.83 mmol) was deprotected by general method E. Following filtration, the solid residue was washed with methanol, which was discarded. The solid residue was washed into a separate flask with water, then freeze dried to give the product as a white solid (0.29 g, 86%). Further purification was performed by Prep HPLC method A. The reaction and the work up were monitored by TLC (eluent: ethanol, R_f_: 0.07). Mp: 200 °C (decomposed); ^1^H-NMR (300 MHz, D_2_O) δ_H_ 5.25 (1H, quart, *J* = 6.9 Hz, 2-H), 3.57 (1H, quart, *J* = 6.9 Hz, 6-H), 1.52 (3H, d, *J* = 7.2 Hz, 3-H), 1.25 (3H, d, *J* = 7.2 Hz, 7-H); ^13^C-NMR (75.5 MHz, D_2_O) δ_C_ 176.2 (5-C), 164.3 (1-C), 49.9 (6-C), 42.0 (2-C), 19.4 (3-C), 19.2 (7-C); ν_max_/cm^–1^ 3247 (N-H), 1643 (C=O) 1561 (N-H); MS(ESI) *m*/*z* 185.1 (M + H)^+^; HRMS (Found (M+H)^+^ 185.1145. Calcd. for C_6_H_13_ON_6_: (M + H)^+^ 185.1145.); (Purity test: HPLC method A).

*Benzyl-N-[(1S)-1-{[(1R)-1-(1H-1,2,3,4-tetrazol-5-yl)ethyl]carbamoyl}ethyl]carbamate* (**13a-LD**) **[18]**: Prepared using general method G. The Fmoc group of Fmoc-1-d-aminoethyltetrazole attached to 2-chlorotrityl chloride resin (resin:loading; 1:1 ratio; 1.2 mmol) was removed, this was monitored by the Fmoc and TNBS tests. After the coupling of Cbz-l-alanine with the resin, the title compound was cleaved off. The cleaving mixture (TFA/DCM/TIS, ratio 5:95:1 eq) was removed by evaporation under reduced pressure, and the residue washed with petroleum ether (40–60 °C) to obtain the product as white crystals (0.34 g, 89%). The reaction and the work up were monitored by TLC (eluent: ethyl acetate and 1 drop of glacial acetic acid, R_f_: 0.07). Mp: 166–168 °C; ^1^H-NMR (300 MHz, DMSO-d_6_) δ_H_ 8.56 (1H, d, *J* = 7.5 Hz, 4-H), 7.35 (6H, m, 8,13-15-H), 5.23 (1H, quint, *J* = 7.2 Hz, 2-H), 5.04 (1H, d, *J* = 12.9 Hz, 11-H_a_), 5.00 (1H, d, *J* = 12.6 Hz, 11-H_b_), 4.10 (1H, quint, *J* = 7.5 Hz, 6-H), 1.50 (3H, d, *J* = 7.2 Hz, 3-H), 1.23 (3H, d, *J* = 6.9 Hz, 7-H); ^13^C-NMR (75.5 MHz, DMSO-d_6_) δ_C_ 172.6 (5-C), 158.0 (1-C), 156.1 (10-C), 137.4 (12-C), 128.8 (14-C), 128.2 (15-C), 128.2 (13-C), 65.9 (11-C), 50.4 (6-C), 40.3 (2-C), 19.6 (3-C), 18.8 (7-C); ν_max_/cm^–1^ 3289 (N-H), 2988, 2874, 2756, 2612 (tetrazole), 1682 (C=O), 1651 (C=O), 1528 (N-H), 1224 (C-O); MS(ESI) *m*/*z* 319.1 (M + H)^+^; CHN [Found: C, 53.10; H, 5.78; N, 26.01. C_14_H_18_N_6_O_3_ requires C, 52.82; H, 5.70; N, 26.40%].

*(2S)-2-Amino-N-[(1R)-1-(1H-1,2,3,4-tetrazol-5-yl)ethyl]propanamide* (**14a-LD**) [18]: Cbz-l-alanyl-1-d-aminoethyltetrazole (**13a-LD**, 0.58 g, 1.83 mmol) was deprotected by general method E. Following filtration, the solid residue was washed with ethanol, which was discarded. The solid residue was washed into a separate flask with water, which was freeze dried and purified by column chromatography (eluent: ethyl acetate – ethanol 4:1, 2:1, 1:1 then 0:1 ratio) to give the pure product as white solid (98 mg, 58%). The reaction and the work up were monitored by TLC (eluent: ethanol, R_f_: 0.13). Mp: 69 °C (phase change), 177 °C (melted); ^1^H-NMR (300 MHz, D_2_O) δ_H_ 5.24 (^1^H, quart, *J* = 6.6 Hz, 2-H), 4.00 (1H, quart, *J* = 6.6 Hz, 6-H), 1.52 (3H, d, *J* = 7.5 Hz, 3-H), 1.48 (3H, d, *J* = 7.8 Hz, 7-H); ^13^C-NMR (75.5 MHz, D_2_O) δ_C_ 170.5 (5-C), 164.0 (1-C), 49.3 (6-C), 42.3 (2-C), 19.1 (3-C), 16.7 (7-C); ν_max_/cm^–1^ 3204 (N-H), 1667 (C=O), 1539 (N-H); MS(ESI) *m*/*z* 185.1 (M + H)^+^; HRMS (Found (M + H)^+^ 185.1145. Calcd. for C_6_H_13_ON_6_: (M + H)^+^ 185.1145.); (Purity test: HPLC method A).

*(S)-Benzyl-(3-((1-(1H-tetrazol-5-yl)ethyl)amino)-3-oxopropyl)carbamate* (**13d-L**): Prepared using general method G. The Fmoc group of Fmoc-1-l-aminoethyltetrazole attached to the 2-chlorotrityl chloride resin (resin:loading; 1:1 ratio; 1.3 mmol) was removed, this was monitored by the Fmoc and TNBS tests. After the coupling of Cbz-β-alanine with the resin, the title compound was cleaved off. The cleaving mixture (TFA/DCM/TIS, ratio 5:95:1 eq) was removed by evaporation under reduced pressure, and the residue washed with petroleum ether (40–60 °C) to obtained the product as white crystals (0.28 g, 67%). The reaction and the work up were monitored by TLC (eluent: ethyl acetate and 1 drop of glacial acetic acid, R_f_: 0.39). Mp: 147–148 °C; ^1^H-NMR (300 MHz, DMSO-d_6_) δ_H_ 16.08 (1H, brs, 9-H), 8.56 (1H, d, *J* = 7.5 Hz, 4-H), 7.35 (5H, m, 13-15-H), 7.21 (1H, m, 8-H), 5.22 (1H, quint, *J* = 7.2 Hz, 2-H), 5.02 (2H, s, 11-H), 3.24 (2H, quart, *J* = 6.9Hz, 7-H), 2.34 (2H, t, *J* = 6.9 Hz, 6-H), 1.48 (3H, d, *J* = 7.2 Hz, 3-H); ^13^C-NMR (75,5 MHz, DMSO-d_6_) δ_C_ 170.6 (5-C), 158.8 (1-C), 156.5 (10-C), 137.6 (12-C), 128.8 (2C, 14-C), 128.2 (15-C), 128.1 (2C, 13-C), 65.7 (11-C), 40.9 (2-C), 37.3 (7-C), 35.9 (6-C), 19.6 (3-C); ν_max_/cm^–1^ 3323 (N-H), 3273 (N-H), 2994, 2882, 2746, 2615 (tetrazole), 1692 (C=O), 1645 (C=O); MS(ESI) (M + Na)^+^ found 341.2 *m*/*z*; HRMS (Found (M + H)^+^ 319.1516. Calcd. for C_14_H_19_O_3_N_6_: (M + H)^+^ 319.1513.); (Purity test: HPLC method C).

*3-Amino-N-[(1S)-1-(1H-1,2,3,4-tetrazol-5-yl)ethyl]propanamide* (**14d-L**): Cbz-β-alanyl-1-l-amino-ethyltetrazole (**13d-L**, 0.20 g, 0.63 mmol) was deprotected by general method E. Following filtration, the solid residue was washed with methanol, which was discarded. The solid residue was washed into a separate flask with water, which was freeze dried to give the product as a light brown hygroscopic amorphous solid (0.09 g, 78%). Further purification was performed by Prep HPLC method A. ^1^H-NMR (300 MHz, D_2_O) δ_H_ 5.21 (1H, quart, *J* = 6.9 Hz, 2-H), 3.17 (2H, m, 7-H), 2.63 (2H, t, *J* = 6.6 Hz, 6-H), 1.48 (3H, d, *J* = 6.9 Hz, 3-H); ^13^C-NMR (75,5 MHz, DMSO-d_6_) δ_C_ 171.6 (5-C), 164.5 (1-C), 40.0 (2-C), 35.8 (7-C), 32.5 (6-C), 19.3 (3-C); ν_max_/cm^–1^ 3232 (N-H), 1641 (C=O); MS(ESI) (M+H)^+^ found 185.1 *m*/*z*; HRMS (Found (M + H)^+^ 185.1145. Calcd. for C_6_H_13_ON_6_: (M + H)^+^ 185.1145.); (Purity test: HPLC method A).

*Benzyl-N-[(1S)-1-{[(1S)-1-(1H-1,2,3,4-tetrazol-5-yl)ethyl]carbamoyl}butyl]carbamate* (**13e-LL**): Prepared using general method G. The Fmoc group of Fmoc-1-l-aminoethyltetrazole attached to 2-chlorotrityl chloride resin (resin:loading; 1:1 ratio; 1.2 mmol) was removed, this was monitored by the Fmoc and TNBS tests. After the coupling of Cbz-L-norvaline with the resin, the title compound was cleaved off. The cleaving mixture (TFA/DCM/TIS, ratio 5:95:1 eq) was removed by evaporation under reduced pressure, and the residue washed with petroleum ether (40–60 °C) to obtain the product as white crystals (0.21, 51%). The reaction and the work up were monitored by TLC (eluent: ethyl acetate and 1 drop of glacial acetic acid, R_f_: 0.2). Mp: 175–178 °C; ^1^H-NMR (300 MHz, DMSO-d_6_) δ_H_ 16.22 (1H, brs, 11-H), 8.55 (1H, d, *J* = 7.5, 4-H), 7.35 (6H, m, 10, 15-17-H), 5.23 (1H, quint, *J* = 7.2 Hz, 2-H), 5.02 (2H, s, 13-H), 4.04 (1H, m, 6-H), 1.59 (2H, m, 7-H), 1.49 (3H, d, *J* = 6.9 Hz, 3-H), 1.27 (2H, m, 8-H), 0.84 (3H, t, *J* = 7.2 Hz, 9-H); ^13^C-NMR (75.5 MHz, DMSO-d_6_) δ_C_ 172.2 (5-C), 159.1 (1-C), 156.4 (12-C), 137.5 (14-C), 128.8 (2C, 16-C), 128.2 (17-C), 128.1 (2C, 15-C), 65.8 (13-C), 54.6 (6-C), 40.9 (2-C), 34.3 (7-C), 19.7 (3-C), 19.1 (8-C), 14.0 (9-C); ν_max_/cm^–1^ 3310 (N-H), 3271 (N-H), 2959, 2877, 2752, 2609 (tetrazole), 1682 (C=O), 1651 (C=O), 1533 (N-H), 1234 (C-O); MS(ESI) *m*/*z* 369.2 (M + Na)^+^; HRMS (Found (M + H)^+^ 347.1830. Calcd. for C_16_H_23_O_3_N_6_: (M + H)^+^ 347.1826.); (Purity test: HPLC method D)

*(S)-N-((S)-1-(1H-tetrazol-5-yl)ethyl)-2-aminopentanamide* (**14e-LL**): Cbz-l-norvalyl-1-l-amino-ethyltetrazole (**13e-LL**, 0.10 g, 0.29 mmol) was deprotected by general method E. Following filtration, the solid residue was washed with methanol, which was discarded. The residue was washed with water, then freeze dried to give the product as a white solid (47 mg, 77%). Further purification was performed by Prep HPLC method A. The reaction and the work up were monitored by TLC (eluent: ethanol, R_f_: 0.16). Mp: 251–254 °C; ^1^H-NMR (300 MHz, D_2_O) δ_H_ 5.28 (1H, quart, *J* = 7.2 Hz, 2-H), 3.79 (1H, t, *J* = 6.6 Hz, 6-H), 1.70 (2H, m, 7-H), 1.54 (3H, d, *J* = 6.9 Hz, 3-H), 1.21 (2H, m, 8-H), 0.81 (3H, t, *J* = 7.2 Hz, 9-H); ^13^C-NMR (75.5 MHz, DMSO-d_6_) δ_C_ 171.2 (5-C), 164.7 (1-C), 53.5 (6-C), 42.0 (2-C), 33.8 (7-C), 19.1 (3-C), 17.6 (8-C), 12.8 (9-C); ν_max_/cm^–1^ 3210 (N-H), 1687 (C=O), 1551 (N-H); MS(ESI) *m*/*z* 213.1 (M + H)^+^; HRMS (Found (M + H)^+^ 213.1458. Calcd. for C_8_H_17_ON_6_: (M + H)^+^ 213.1460.); (Purity test: HPLC method A).

*Benzyl-N-[(1S)-3-methyl-1-{[(1S)-1-(1H-1,2,3,4-tetrazol-5-yl)ethyl]carbamoyl}butyl]carbamate* (**13h-LL**): Prepared using general method G. The Fmoc group of Fmoc-1-l-aminoethyltetrazole attached to 2-chlorotrityl chloride resin (resin:loading; 1:2 ratio; 1.5 mmol) was removed, this was monitored by the Fmoc and TNBS tests. After the coupling of Cbz-l-leucine with the resin, the title compound was cleaved off. The cleaving mixture (TFA/DCM/TIS, ratio 5:95:1 eq) was removed by evaporation under reduced pressure, and the residue washed with petroleum ether (40–60 °C) to obtain the product as white crystals (0.45, 84%). The reaction and the work up were monitored by TLC (eluent: ethyl acetate – ethanol – drop of glacial acetic acid 1:1 ratio, R_f_: 0.64). Mp: 149–150 °C; ^1^H-NMR (300 MHz, DMSO-d_6_) δ_H_ 16.1 (1H, brs, 11-H), 8.55 (1H, d, *J* = 7.5 Hz, 4-H), 7.36 (6H, m, 10,15,16,17-H), 5.22 (1H, quint, *J* = 6.9 Hz, 2-H), 5.04 (1H, d, *J* = 12.6, 13a-H), 5.00 (1H, d, *J* = 12.6 Hz, 13a-H), 4.07 (1H, m, 6-H), 1.60 (1H, m, 8-H), 1.49 (3H, d, *J* = 7.2 Hz, 3-H), 1.42 (2H, m, 7-H), 0.85 (3H, d, *J* = 6.6 Hz, 9a-H), 0.84 (3H, d, *J* = 6.6 Hz, 9b-H); ^13^C-NMR (75.5 MHz, DMSO-d_6_) δ_C_ 172.6 (5-C), 156.4 (12-C), 137.5 (14-C), 128.8 (16-C), 128.2 (17-C), 128.1 (15-C), 65.8 (13-C), 53.3 (6-C), 41.0 (2-C), 40.3 (7-C), 24.6 (8-C), 23.6 (9a-C), 21.8 (9b-C), 19.6 (3-C); ν_max_/cm^–1^ 3316 (N-H), 3271 (N-H), 2961, 2876, 2743, 2615 (tetrazole), 1682 (C=O), 1653 (C=O), 1530 (N-H), 1233 (C-O); MS(ESI) *m*/*z* 360.8 (M + H)^+^; CHN [Found: C, 56.82.; H, 6.68; N, 23.05. C_17_H_24_N_6_O_3_ requires C, 56.65; H, 6.71; N, 23.32%].

*(2S)-2-Amino-4-methyl-N-[(1S)-1-(1H-1,2,3,4-tetrazol-5-yl)ethyl]pentanamide* (**14h-LL**): Cbz-l-leucyl-1-l-aminoethyltetrazole (**13h-LL**, 0.95 g, 2.6 mmol) was deprotected by general method E. Following filtration, the solid residue was washed with ethanol, which was discarded. The residue was washed with water, then freeze dried and washed with diethyl ether to give the product as a white solid (0.48 g, 81%). Further purification was performed by Prep HPLC method A. The reaction and the work up were monitored by TLC (eluent: ethanol, R_f_: 0.3). Mp: 210 °C (decomposed); ^1^H-NMR (300 MHz, D_2_O) δ_H_ 5.28 (1H, quart, *J* = 7.2 Hz, 2-H), 3.49 (1H, t, *J* = 7.2Hz, 6-H), 1.53 (3H, d, *J* = 7.2 Hz, 3-H), 1.44 (3H, m, 7,8-H), 0.82 (3H, d, *J* = 4.8 Hz, 9a-H), 0.80 (3H, d, *J* = 5.1 Hz, 9b-H); ^13^C-NMR (75.5 MHz, DMSO-d_6_) δ_C_ 175.6 (5-C), 169.7 (1-C), 52.9 (6-C), 42.7 (7-C), 41.8 (2-C), 24.0 (8-C), 21.9 (9a-C), 21.4 (9b-C), 16.8 (3-C); ν_max_/cm^–1^ 3219 (N-H), 1653 (C=O), 1562 (N-H); MS(ESI) *m*/*z* 227.2 (M + H)^+^; HRMS (Found (M + Na)^+^ 249.1436. Calcd. for C_9_H_18_ON_6_Na: (M + Na)^+^ 249.1434.); (Purity test: HPLC method A).

*Benzyl-N-[(1S)-3-methyl-1-{[(1R)-1-(1H-1,2,3,4-tetrazol-5-yl)ethyl]carbamoyl}butyl]carbamate* (**13h-LD**): Prepared using general method G. The Fmoc group of Fmoc-1-d-aminoethyltetrazole attached to 2-chlorotrityl chloride resin (resin:loading; 1:1 ratio; 1.2 mmol) was removed, this was monitored by Fmoc and TNBS tests. After the coupling of Cbz-l-leucine with the resin, the title compound was cleaved off. The cleaving mixture (TFA/DCM/TIS, ratio 5:95:1 eq) was removed by evaporation under reduced pressure, and washed with petroleum ether (40–60 °C) to obtain the product as white crystals (0.30 g, 69%). The reaction and the work up were monitored by TLC (eluent: ethyl acetate – ethanol 1:1 ratio, R_f_: 0.63). Mp: 143–144 °C; ^1^H-NMR (300 MHz, DMSO-d_6_) δ_H_ 16.2 (1H, brs, 11-H), 8.61 (1H, d, *J* = 7.5 Hz, 4-H), 7.35 (6H, m, 10,15,16,17-H), 5.26 (1H, quint, *J* = 7.5 Hz, 2-H), 5.04 (1H, d, *J* = 13.2, 13a-H), 5.00 (1H, d, *J* = 12.9 Hz, 13a-H), 4.11 (1H, m, 6-H), 1.61 (1H, m, 8-H), 1.45 (3H, d, *J* = 6.9 Hz, 3-H), 1.42 (2H, m, 7-H), 0.88 (3H, d, *J* = 6.0 Hz, 9a-H), 0.86 (3H, d, *J* = 5.7 Hz, 9b-H); ^13^C-NMR (75.5 MHz, DMSO-d_6_) δ_C_ 172.4 (5-C), 158.5 (1-C), 156.4 (12-C), 137.5 (14-C), 128.8 (16-C), 128.2 (17-C), 128.1 (15-C), 65.9 (13-C), 53.4 (6-C), 41.4 (2-C), 40.3 (7-C), 24.7 (8-C), 23.5 (9a-C), 21.9 (9b-C), 19.6 (3-C); ν_max_/cm^–1^ 3312 (N-H), 3269 (N-H), 2959, 2876, 2756, 2621 (tetrazole) 1690 (C=O), 1649 (C=), 1528 (N-H), 1234 (C-O); MS(ESI) *m*/*z* 361.2 (M + H); HRMS (Found (M + H)^+^ 361.1987. Calcd. for C_17_H_25_O_3_N_6_: (M + H)^+^ 361.1983.); (Purity test: HPLC method D).

*(2S)-2-Amino-4-methyl-N-[(1R)-1-(1H-1,2,3,4-tetrazol-5-yl)ethyl]pentanamide* (**14h-LD**): Cbz-l-leucyl-1-d-aminoethyltetrazole (**13h-LD**, 0.25 g, 0.7 mmol) was deprotected by general method E. Following filtration, the solid residue was washed with ethanol, which was discarded. The residue was washed with water, which was freeze dried and purified by column chromatography (eluent: ethyl acetate – ethanol 1:1 – 1:2 – 0:1) to give the pure product as a white solid (83 mg, 53%). The reaction and the work up were monitored by TLC (eluent: ethanol, R_f_: 0.21). Mp: 136 °C (decomposed), 161 °C (melted); ^1^H-NMR (300 MHz, D_2_O) δ_H_ 5.27 (1H, quart, *J* = 6.9 Hz, 2-H), 3.95 (1H, t, *J* = 6.9Hz, 6-H), 1.71 (3H, m, 7,8-H), 1.53 (3H, d, *J* = 7.2 Hz, 3-H), 0.93 (3H, d, *J* = 6.0 Hz, 9a-H), 0.91 (3H, d, *J* = 6.0 Hz, 9b-H); ^13^C-NMR (75.5 MHz, DMSO-d_6_) δ_C_ 169.8 (5-C), 164.7 (1-C), 52.2 (6-C), 42.4 (2-C), 39.9 (7-C), 24.0 (8-C), 21.6 (9a-C), 21.3 (9b-C), 19.1 (3-C); ν_max_/cm^–1^ 3194 (N-H), 1667 (C=O), 1543 (N-H); MS(ESI) *m*/*z* 227.2 (M + H)^+^; HRMS (Found (M + H)^+^ 227.1625 Calcd. for C_9_H_19_ON_6_: (M + H)^+^ 227.1620.); (Purity test: HPLC method A).

*Benzyl-N-[(1S,2S)-2-methyl-1-{[(1S)-1-(1H-1,2,3,4-tetrazol-5-yl)ethyl]carbamoyl}butyl]carbamate* (**13i-LL**): Prepared using general method G. The Fmoc group of Fmoc-1-l-aminoethyltetrazole attached to 2-chlorotrityl chloride resin (resin:loading; 1:1 ratio; 1.8 mmol) was removed, this was monitored by the Fmoc and TNBS tests. After the coupling of Cbz-L-isoleucine with the resin, the title compound was cleaved off. The cleaving mixture (TFA/DCM/TIS, ratio 5:95:1 eq) was removed by evaporation under reduced pressure, and washed with petroleum ether (40–60 °C) to obtain the product as white crystals (0.40 g, 62%). The reaction and the work up were monitored by TLC (eluent: ethyl acetate – ethanol 10:1 ratio, R_f_: 0.13). Mp: 186–189 °C; ^1^H-NMR (300 MHz, DMSO-d_6_) δ_H_ 8.58 (1H, d, *J* = 7.2 Hz, 4-H), 7.35 (5H, m, 16,17,18-H), 7.24 (1H, d, *J* = 9.0 Hz, 11-H), 5.24 (1H, quint, *J* = 6.9 Hz, 2-H), 5.05 (1H, d, *J* = 12.6 Hz, 14a-H), 5.00 (1H, d, *J* = 12.6 Hz, 14b-H), 3.92 (1H, t, *J* = 8.1 Hz, 6-H), 1.70 (1H, m, 7-H), 1.50 (3H, d, *J* = 6.9 Hz, 3-H), 1.35 (1H, m, 9a-H), 1.07 (1H, m, 9b-H), 0.77 (3H, t, *J* = 7.5 Hz, 10-H), 0.75 (3H, d, *J* = 7.5 Hz, 8-H); ^13^C-NMR (75.5 MHz, DMSO-d_6_) δ_C_ 171.4 (5-C), 158.3 (1-C), 156.5 (13-C), 137.5 (15-C), 128.8 (17-C), 128.2 (18-C), 128.1 (16-C), 65.9 (14-C), 59.3 (6-C), 39.9 (2-C), 36.9 (7-C), 24.7 (9-C), 19.5 (3-C), 15.7 (8-C), 11.3 (10-C); ν_max_/cm^–1^ 3279 (N-H), 2963, 2878, 2693, 2608 (tetrazole), 1690 (C=O), 1651 (C=O), 1535 (N-H), 1234 (C-O); MS(ESI) *m*/*z* 361.1 (M + H)^+^; CHN [Found: C, 56.61.; H, 6.77; N, 23.01. C_17_H_24_N_6_O_3_ requires C, 56.65; H, 6.71; N, 23.32%].

*(2S,3S)-2-amino-3-methyl-N-[(1S)-1-(1H-1,2,3,4-tetrazol-5-yl)ethyl]pentanamide* (**14i-LL**): Cbz-l-isoleucyl-1-l-aminoethyltetrazole (**13i-LL**, 0.35 g, 0.9 mmol) was deprotected by general method E. Following filtration, the solid residue was washed with ethanol, which was discarded. The residue was washed with water, which was freeze dried to give the product as white solid (190 mg, 86%). Due to hygroscopicity, the corresponding hydrochloride salt was made with 2M HCl in diethyl ether solution (20 mL) for characterization. Further purification was performed by Prep HPLC method A. The reaction and the work up were monitored by TLC (eluent: ethanol, R_f_: 0.31). Mp: 164 - 167 °C (melted and decomposed); ^1^H-NMR (300 MHz, D_2_O) δ_H_ 5.39 (1H, quart, *J* = 7.2 Hz, 2-H), 3.85 (1H, d, *J* = 5.7 Hz, 6-H), 1.92 (1H, m, 7-H), 1.65 (3H, d, *J* = 7.2 Hz, 3-H), 1.31 (1H, m, 9a-H), 1.14 (1H, m, 9b-H), 0.89 (3H, d, *J* = 6.9 Hz, 8-H), 0.83 (3H, t, *J* = 7.5 Hz, 10-H); ^13^C-NMR (75.5 MHz, DMSO-d_6_) δ_C_ 169.1 (5-C), 158.1 (1-C), 57.7 (6-C), 40.7 (2-C), 36.3 (7-C), 23.9 (9-C), 17.6 (3-C), 14.1 (8-C), 10.4 (10-C); ν_max_/cm^–1^ 3240 (N-H), 1682 (C=O), 1574 (N-H); MS(ESI) *m*/*z* 227.2 (M + H)^+^; HRMS (Found (M + Na)^+^ 249.1436. Calcd. for C_9_H_18_ON_6_Na: (M + Na)^+^ 249.1434.); (Purity test: HPLC method A).

*Benzyl-(5S)-2-oxo-5-{[(1S)-1-(1H-1,2,3,4-tetrazol-5-yl)ethyl]carbamoyl}pyrrolidine-1-carboxylate* (**13f-LL**): Prepared using general method G. The Fmoc group of the Fmoc-1-l-aminoethyltetrazole attached to 2-chlorotrityl chloride resin (resin:loading; 1:2 ratio; 2.0 mmol) was removed. After coupling of Cbz-L-pyroglutamine with the resin, the title compound was cleaved off. The cleaving mixture (TFA/DCM/TIS, ratio 5:95:1 eq) was subsequently removed by evaporation under reduced pressure. The crude product was purified by column chromatography (eluent: ethyl acetate – ethanol 10:1, 6:1 then 1:1 ratio) to obtain the product as white crystals (0.46g, 65%). The reaction and the work up were monitored by TLC (eluent: ethyl acetate – ethanol 10:1, R_f_: 0.09). Mp: 195 °C (decomposed); ^1^H-NMR (300 MHz, DMSO-d_6_) δ_H_ 16.05 (1H, brs, 11-H), 8.93 (1H, d, *J* = 7.5 Hz, 4-H), 7.36 (5H, m, 15-17-H), 5.17 (3H, m, 2,13-H), 4.64 (1H, dd, *J* = 2.7 Hz, *J* = 9.0 Hz, 6-H), 2.41 (2H, m, 8-H), 2.26 (1H, m, 7-H_a_), 1.94 (1H, m, 7-H_b_), 1.42 (3H, d, *J* = 6.9 Hz, 3-H); ^13^C-NMR (75.5 MHz, DMSO-d_6_) δ_C_ 174.1 (9-C), 171.0 (5-C), 158.3 (1-C), 150.9 (12-C), 136.0 (14-C), 128.8 (2C, 16-C), 128.5 (17-C), 128.0 (15-C), 67.5 (13-C), 59.3 (6-C), 40.2 (2-C), 31.3 (8-C), 22.3 (7-C), 19.6 (3-C); ν_max_/cm^–1^ 3289 (N-H), 2993, 2868, 2739, 2615 (tetrazole), 1780 (C=O), 1701 (C=O), 1655 (C=O), 1545 (N-H), 1236 (C-O); MS(ESI) *m*/*z* 381.2 (M + Na)^+^. CHN [Found: C, 53.63; H, 5.10; N, 23.18. C_16_H_18_N_6_O_4_ requires C, 53.63; H, 5.06; N, 23.45%].

*(S)-N-((S)-1-(1H-tetrazol-5-yl)ethyl)-5-oxopyrrolidine-2-carboxamide* (**14f-LL**): Cbz-l-pyroglutamyl-1-l-aminoethyltetrazole (**13f-LL**, 0.44 g, 1.2 mmol) was reacted by general method E. Following filtration, the solid residue was washed with methanol and the filtrate was evaporated under reduced pressure. The residual oil was triturated with diethyl ether to give the title compound as a white solid (0.23 g, 84%). Further purification was performed by Prep HPLC method A. The reaction and the work up were monitored by TLC (ethanol, R_f_: 0.3). Mp: 160 °C (decomposed); ^1^H-NMR (300 MHz, D_2_O) δ_H_ 5.35 (1H, quart, *J* = 7.2 Hz, 2-H), 4.34 (1H, dd, *J* = 4.8 Hz, *J* = 8.4 Hz, 6-H), 2.52 (1H, m, 7-H_a_), 2.40 (2H, m, 8-H) 2.04 (1H, m, 7-H_b_), 1.61 (3H, d, *J* = 6.9 Hz, 3-H); ^13^C-NMR (75.5 MHz, D_2_O) δ_C_ 182.3 (9-C), 174.6 (5-C), 159.0 (1-C), 56.9 (6-C), 41.0 (2-C), 29.3 (8-C), 25.0 (7-C), 18.1 (3-C); ν_max_/cm^–1^ 3262 (N-H), 1649 (C=O), 1547 (N-H); MS(ESI) *m*/*z* 225.1 (M + H)^+^. HRMS (Found (M + H)^+^ 225.1096. Calcd. for C_8_H_13_O_2_N_6_: (M + H)^+^ 225.1095.); (Purity test: HPLC method A).

*Benzyl-N-[(1S)-2-phenyl-1-{[(1S)-1-(1H-1,2,3,4-tetrazol-5-yl)ethyl]carbamoyl}ethyl]carbamate* (**13j-LL**): Prepared using general method G. The Fmoc group of Fmoc-1-l-aminoethyltetrazole attached to 2-chlorotrityl chloride resin (resin:loading; 1:1 ratio; 1.0 mmol) was removed, this was monitored by the Fmoc and TNBS tests. After the coupling of Cbz-l-phenylalanine with the resin, the title compound was cleaved off. The cleaving mixture (TFA/DCM/TIS, ratio 5:95:1 eq) was removed by evaporation under reduced pressure, and washed with petroleum ether (40–60 °C) to obtain the product as white crystals (0.32 g, 79%). Further purification was performed using a CombiFlash EZ Prep system (Teledyne-ISCO, Lincoln, NE, USA, eluent: gradient: dichloromethane – methanol 0–30%). The reaction and the work up were monitored by TLC (eluent: ethyl acetate, R_f_: 0.11). Mp: 191–193 °C; ^1^H-NMR (300 MHz, DMSO-d_6_) δ_H_ 16.21 (1H, brs, 8.72 (1H, d, *J* = 7.5 Hz, 4-H), 7.44 (1H, d, *J* = 8.7 Hz, 12-H), 7.33 - 7.24 (10H, m, 9,10,11,17,18,19-H), 5.27 (1H, quint, 6.9 Hz, 2-H), 4.96 (1H, d, *J* = 13.5 Hz, 15a-H), 4.92 (1H, d, *J* = 13.5 Hz, 15b-H), 4.30 (1H, m, 6-H), 3.03 (1H, dd, *J* = 3.3 Hz, *J* = 13.8 Hz, 7a-H), 2.72 (1H, dd, *J* = 10.8 Hz, *J* = 13.5 Hz, 7b-H), 1.52 (3H, d, *J* = 6.9 Hz, 3-H); ^13^C-NMR (75.5 MHz, DMSO-d_6_) δ_C_ 171.7 (5-C), 158.7 (1-C), 156.3 (14-C), 138.5 (8-C), 137.5 (16-C), 129.6, 128.7, 128.5, 128.1, 127.9, 126.7 (9,10,11,17,18,19-C), 65.7 (15-C), 56.5 (6-C), 40.0 (2-C), 37.8 (7-C), 19.7 (3-C); ν_max_/cm^–1^ 3300 (N-H), 2984, 2874, 2741, 2615 (tetrazole), 1684 (C=O), 1651 (C=O), 1531 (N-H), 1234 (C-O); MS(ESI) *m*/*z* 417.2 (M + Na)^+^; CHN [Found: C, 60.78; H, 5.68; N, 20.92. C_20_H_22_N_6_O_3_ requires C, 60.90; H, 5.62; N, 21.31%].

*(2S)-2-Amino-3-phenyl-N-[(1S)-1-(1H-1,2,3,4-tetrazol-5-yl)ethyl]propanamide* (**14j-LL**): Cbz-l-phenylalanyl-1-l-aminoethyltetrazole (**13j-LL**, 0.30 g, 0.8 mmol) was deprotected by general method E. Following filtration, the solid residue was washed with ethanol, which was discarded. The residue was washed with water, which was freeze dried, then purified by column chromatography (eluent: dichloromethane – ethanol 10:1, 8:1, 6:1, 1:1 then 0:1 ratio) to give the pure product as a white solid (40 mg, 20%). Further purification was performed by Prep HPLC method A. The reaction and the work up were monitored by TLC (eluent: ethanol, R_f_: 0.39). Mp: 152–155 °C; ^1^H-NMR (300 MHz, D_2_O) δ_H_ 7.15 (3H, m, 10,11-H), 7.01 (2H, m, 9-H), 5.24 (1H, quart, *J* = 6.9 Hz, 2-H), 3.77 (1H, m, 6-H), 2.97 (1H, dd, *J* = 6.0, *J* = 13.5 Hz, 7a-H), 2.89 (1H, dd, *J* = 7.8, *J* = 13.5 Hz, 7b-H), 1.43 (3H, d, *J* = 7.2 Hz, 3-H); ^13^C-NMR (75.5 MHz, D_2_O) δ_C_ 163.5 (1-C), 150.6 (5-C), 135.5 (8-C), 129.2 (9-C), 128.6 (10-C), 127.1 (11-C), 55.5 (6-C), 41.3 (2-C), 39.3 (7-C), 19.3 (3-C); ν_max_/cm^–1^ 3217 (N-H), 1651 (C=O), 1528 (N-H); MS(ESI) *m*/*z* 283.2 (M + Na)^+^. HRMS (Found (M + H)^+^ 261.1462. Calcd. for C_12_H_17_ON_6_: (M + H)^+^ 261.1458.); (Purity test: HPLC method A).

*Benzyl-N-[(1S)-1-{[(1S)-1-(1H-1,2,3,4-tetrazol-5-yl)propyl]carbamoyl}ethyl]carbamate* (**13b-LL**): Prepared using general method G. The Fmoc group of the Fmoc-1-l-aminopropyltetrazole attached to 2-chlorotrityl chloride resin (resin:loading; 1:1 ratio; 2.9 mmol) was removed. After coupling of Cbz-l-alanine with the resin, the title compound was cleaved off. The cleaving mixture (TFA/DCM/TIS, ratio 5:95:1 eq) was removed by evaporation under reduced pressure. The crude product was subsequently washed with petroleum ether (40–60 °C) to obtain the product as white crystals (0.49 g, 52%). Mp: 174–175 °C; ^1^H-NMR (300 MHz, DMSO-d_6_) δ_H_ 8.44 (1H, d, *J* = 7.5 Hz, 5-H), 7.41 (1H, d, *J* = 7.2 Hz, 9-H), 7.34 (5H, m, 14,15,16-H), 5.06 (1H, m, 2-H), 5.02 (2H, s, 12-H), 4.10 (1H, quint, *J* = 6.9 Hz, 7-H), 1.88 (2H, m, 3-H), 1.19 (3H, d, *J* = 7.2 Hz, 8-H), 0.87 (3H, t, *J* = 7.2 Hz, 4-H); ^13^C-NMR (75.5 MHz, DMSO-d_6_) δ_C_ 173.0 (6-C), 156.8 (1-C), 156.1 (11-C), 137.5 (13-C), 128.8 (15-C), 128.2 (16-C), 128.1 (14-C), 65.8 (12-C), 50.4 (7-C), 45.8 (2-C), 26.8 (3-C), 18.4 (8-C), 10.6 (4-C); ν_max_/cm^–1^ 3300 (N-H), 3265 (N-H), 2976, 2876, 2735, 2617 (tetrazole), 1691 (C=O), 1651 (C=O), 1533 (N-H), 1246 (C-O); MS(ESI) *m*/*z* 333.1 (M + H)^+^; CHN [Found: C, 54.20; H, 6.11; N, 24.91. C_15_H_20_N_6_O_3_ requires C, 54.21; H, 6.07; N, 25.29%].

*(2S)-2-Amino-N-[(1S)-1-(1H-1,2,3,4-tetrazol-5-yl)propyl]propanamide* (**14b-LL**): Cbz-l-alanyl-1-l-aminopropyltetrazole (**13b-LL**, 0.40 g, 1.2 mmol) was deprotected by general method E. Following filtration, the solid residue was washed with ethanol, which was discarded. The residue was washed with water, which was freeze dried to give the product as a white solid (206 mg, 86%). Due to poor solubility, the corresponding hydrochloride salt was prepared using 2M HCl in diethyl ether solution (20 mL) for characterization (hygroscopic solid). Further purification was performed by Prep HPLC method A. The reaction and the work up were monitored by TLC (eluent: ethanol, R_f_: 0.12). ^1^H-NMR (300 MHz, DMSO-d_6_) δ_H_ 9.17 (1H, d, *J* = 7.6 Hz, 5-H), 8.34 (3H, brs, 9-H), 5.09 (1H, quart, *J* = 7.8 Hz, 2-H), 3.93 (1H, quart, *J* = 6.7 Hz, 7-H), 1.92 (2H, m, 3-H), 1.38 (3H, d, *J* = 7.0 Hz, 8-H), 0.90 (3H, t, *J* = 7.4 Hz, 4-H); ^13^C-NMR (75.5 MHz, DMSO-d_6_) δ_C_ 170.0 (6-C), 158.0 (1-C), 48.6 (7-C), 46.4 (2-C), 26.8 (3-C), 17.4 (8-C), 10.6 (4-C); IR; MS(ESI) *m*/*z* 199.1 (M + H)^+^; HRMS (Found (M + H)^+^ 199.1302. Calcd. for C_7_H_15_ON_6_: (M + H)^+^ 199.1302.); (Purity test: HPLC method A).

*N-[(1S)-1-{[(1S)-1-{[(1S)-1-(1H-1,2,3,4-tetrazol-5-yl)ethyl]carbamoyl}ethyl]carbamoyl} ethyl]carbamate* (**13-LLL**): Prepared using general method G. The Fmoc group of Fmoc-1-l-aminoethyltetrazole attached to the 2-chlorotrityl chloride resin (resin:loading; 1:1 ratio; 1.0 mmol) was removed, this was monitored by the Fmoc and TNBS tests. After the coupling of Fmoc-l-alanine with the resin, the Fmoc group was removed again. Finally, Cbz-l-alanine was attached to the resin, and the title compound was cleaved off. The cleaving mixture (TFA/DCM/TIS, ratio 5:95:1 eq) was removed by evaporation under reduced pressure, and washed with petroleum ether (40–60 °C) to obtain the product as white crystals (0.19 g, 47%). Further purification was performed by Prep HPLC method A. Mp: 199–201 °C (melted and decomposed); ^1^H-NMR (300 MHz, DMSO-d_6_) δ_H_ 16.09 (13-H), 8.48 (1H, d, *J* = 6.9 Hz, 4-H), 7.93 (1H, d, *J* = 7.5 Hz, 8-H), 7.42 (1H, d, *J* = 7.2 Hz, 12-H), 7.35 (5H, m, 17-19-H), 5.21 (1H, quint, *J* = 6.9 Hz, 2-H), 5.02 (2H, s, 15-H), 4.29 (1H, quint, *J* = 7.2 Hz, 6-H), 4.06 (1H, quint, *J* = 7.2 Hz, 10-H), 1.49 (3H, d, *J* = 6.9 Hz, 3-H), 1.19 (6H, d, *J* = 6.9 Hz, 7,11-H); ^13^C-NMR (75.5 MHz, DMSO-d_6_) δ_C_ 172.6 (9-C), 172.3 (5-C), 159.1 (1-C), 156.1 (14-C), 137.5 (16-C), 128.8 (18-C), 128.2 (19-C), 128.1 (17-C), 65.8 (15-C), 50.4 (10-C), 48.4 (6-C), 40.9 (2-C), 19.7 (3-C), 18.5 (7-C), 18.4 (11-C); ν_max_/cm^–1^ 3298(N-H), 1694 (C=O), 1643 (C=O), 1526 (N-H), 1219 (C-O); MS(ESI) *m*/*z* 390.2 (M + H)^+^; HRMS (Found (M + H)^+^ 390.1883. Calcd. for C_17_H_24_O_4_N_7_: (M + H)^+^ 390.1884.); (Purity test: HPLC method C).

*(S)-N-((S)-1-(1H-tetrazol-5-yl)ethyl)-2-((S)-2-aminopropanamido)propanamide* (**14c-LLL**): Cbz-l-alanyl-l-alanyl-1-l-aminoethyltetrazole (**13c-LLL**, 0.12 g, 0.31 mmol) was reacted by the general method E. Following the filtration, the solid residue was washed with methanol, then the filtrate was evaporated under reduced pressure to give the crude product (1.11 g, 95%), further recrystallization from a methanol - ethyl acetate mixture provided the pure product as white crystals (0.477 g, 40%). Further purification was performed by Prep HPLC method A. The reaction and the work up were monitored by TLC (eluent: ethanol, R_f_: 0.13). Mp: 135 °C (melted and decomposed); ^1^H-NMR (300 MHz, D_2_O) δ_H_ 5.25 (1H, quart, *J* = 7.2 Hz, 2-H), 4.30 (1H, quart, *J* = 7.2 Hz, 6-H), 3.95 (1H, quart, *J* = 6.9 Hz, 10-H), 1.52 (3H, d, *J* = 7.2 Hz, 3-H), 1.43 (3H, d, *J* = 7.2 Hz, 11-H), 1.33 (3H, d, *J* = 7.2 Hz, 7-H); ^13^C-NMR (75.5 MHz, D_2_O) δ_C_ 173.6 (5-C), 171.7 (9-C), 164.1 (1-C), 49.8 (6-C), 49.0 (10-C), 42.0 (2-C), 19.4 (3-C), 17.0 (11-C), 16.5 (7-C); ν_max_/cm^–1^ 3252 (N-H), 1639 (C=O), 1539 (N-H); MS(ESI) *m*/*z* 256.1 (M + H)^+^; HRMS (Found (M + H)^+^ 256.1520. Calcd. for C_9_H_18_O_2_N_7_: (M + H)^+^ 256.1516.); (Purity test: HPLC method A).

*3-{[(1S)-1-{[(1S)-1-(1H-1,2,3,4-tetrazol-5-yl)ethyl]carbamoyl}ethyl]carbamoyl}propanoic acid* (**16-LL**): Prepared using general method G. The Fmoc group of Fmoc-1-l-aminoethyltetrazole attached to the 2-chlorotrityl chloride resin (resin:loading; 1:2 ratio; 1.4 mmol) was removed. After the coupling of Fmoc-L-alanine with the resin, the Fmoc group was removed again. Finally, a succinyl group was attached to the resin using succinic anhydride, and the title compound was cleaved off. The cleaving mixture (TFA/water/TIS, ratio 95:2.5:2.5) was removed by evaporation under reduced pressure, and recrystallized from a water – acetone solvent mixture to obtain the product as white crystals (0.16 g, 40%). The reaction and the work up were monitored by TLC (eluent: ethyl acetate – ethanol – drop of glacial acetic acid 1:1 ratio, R_f_: 0.21). Mp: 170–171 °C; ^1^H-NMR (300 MHz, D_2_O) δ_H_ 5.32 (1H, quart, *J* = 7.2 Hz, 2-H), 4.28 (1H, quart, *J* = 7.2 Hz, 6-H), 2.64 (2H, m, 11-H), 2.54 (2H, m, 10-H), 1.62 (3H, d, *J* = 6.9 Hz, 3-H), 1.35 (3H, d, *J* = 7.2 Hz, 7-H); ^13^C-NMR (75.5 MHz, D_2_O) δ_C_ 176.9 (12-C), 175.0 (9-C), 174.9 (5-C), 158.5 (1-C), 49.7 (6-C), 40.7 (2-C), 29.9 (10-C), 29.0 (11-C), 17.8 (3-C), 16.4 (7-C); ν_max_/cm^–1^ 3358 (N-H), 3254 (O-H), 2992, 2945, 2886, 2778, 2712 (tetrazole), 1713 (C=O), 1636 (C=O), 1543 (N-H), 1526 (N-H), 1231 (C-O); MS(MALDI) *m*/*z* 285.30 (M + H)^+^; HRMS (Found (M + H)^+^ 285.1310. Calcd. for C_10_H_17_O_4_N_6_: (M + H)^+^ 285.1306.); (Purity test: HPLC method B).

*3-{[(1S)-1-{[(1S)-1-{[(1S)-1-(1H-1,2,3,4-tetrazol-5-yl)ethyl]carbamoyl}ethyl]carbamoyl}ethyl]-carbamoyl}propanoic acid* (**17-LLL**): Prepared using general method G. The Fmoc group of Fmoc-1-l-aminoethyltetrazole attached to the 2-chlorotrityl chloride resin (resin:loading; 1:2 ratio; 2.8 mmol) was removed. After the coupling of Fmoc-l-alanine with the resin, the Fmoc group was removed, and the free amino group was coupled with Fmoc-l-alanine again. After removing the Fmoc protection, a succinyl group was attached to the resin using succinic anhydride, and the title compound was cleaved off. The cleaving mixture (TFA/water/TIS, ratio 95:2.5:2.5) was removed by evaporation under reduced pressure, and triturated with diethyl ether to obtain the product as white crystals (0.39 g, 79%). Further purification was performed by Prep HPLC method A. The reaction and the work up were monitored by TLC (eluent: ethyl acetate – ethanol – drop of glacial acetic acid 1:1 ratio, R_f_: 0.22). Mp: 186 °C (melted and decomposed); ^1^H-NMR (300 MHz, D_2_O) δ_H_ 5.32 (1H, quart, *J* = 7.2 Hz, 2-H), 4.29 (1H, quart, *J* = 7.2 Hz, 6-H), 4.21 (1H, quart, *J* = 7.2 Hz, 10-H), 2.66 (2H, m, 16-H), 2.55 (2H, m, 15-H), 1.62 (3H, d, *J* = 6.9 Hz, 3-H), 1.35 (3H, d, *J* = 7.2 Hz, 7-H), 1.34 (3H, d, *J* = 7.2 Hz, 11-H); ^13^C-NMR (75.5 MHz, D_2_O) δ_C_ 177.0 (17-C), 175.3 (14-C), 175.2 (9-C), 174.6 (5-C), 158.1 (1-C), 50.0 (10-C), 49.6 (6-C), 40.7 (2-C), 29.8 (16-C), 29.0 (15-C), 17.8 (3-C), 16.3 (7-C), 16.2 (11-C); ν_max_/cm^–1^ 3343 (N-H), 3381 (O-H), 1707 (C=O), 1651 (C=O), 1634 (C=O), 1539 (N-H), 1201 (C-O); MS(MALDI) *m*/*z* 356.30 (M + H)^+^. HRMS (Found (M + H)^+^ 356.1682. Calcd. for C_13_H_22_O_5_N_7_: (M + H)^+^ 356.1677.) (Purity test: HPLC method A).

*3-{[(1S)-1-{[(1S)-1-{[(1S)-1-{[(1S)-1-(1H-1,2,3,4-tetrazol-5-yl)ethyl]carbamoyl}ethyl]carbamoyl}ethyl]-carbamoyl}ethyl]carbamoyl}propanoic acid* (**18-LLLL**): Prepared using general method G. The Fmoc group of Fmoc-1-l-aminoethyltetrazole attached to the 2-chlorotrityl chloride resin (resin:loading; 1:1 ratio; 1.5 mmol) was removed, and resin was coupled with Fmoc-l-alanine, the Fmoc deprotection and coupling with Fmoc-l-alanine procedure was repeated 2 more times. After removing the Fmoc protection for the final time, a succinyl group was attached to the resin using succinic anhydride, and the title compound was cleaved off. The cleaving mixture (TFA/water/TIS, ratio 95:2.5:2.5) was removed by evaporation under reduced pressure. The residue was triturated with diethyl ether to obtain the product as a white solid (0.44 g, 68%). Further purification was carried out by column chromatography (eluent: ethyl acetate – ethanol 4:1, 1:1 then 0:1 ratio) and by Prep HPLC method A. The reaction and the work up were monitored by TLC (eluent: ethanol, R_f_: 0.21). Mp: 231 °C (decomposed); ^1^H-NMR (300 MHz, D_2_O) δ_H_ 5.32 (1H, quart, J = 7.0 Hz, 2-H), 4.26 (3H, m, 6,10,14-H), 2.64 (2H, m, 19-H), 2.58 (2H, m, 18-H), 1.63 (3H, d, J = 7.0 Hz, 3-H), 1.36 (9H, d, J = 7.2 Hz, 7,11,15-H); ^13^C-NMR (75.5 MHz, D_2_O) δ_C_ 177.9 (20-C), 175.5 (17-C), 175.3, 174.9, 174.4 (5,9,13-C), 158.7 (1-C), 50.1, 49.8, 49.6 (6,10,14-C), 40.9 (2-C), 29.9 (18-H), 29.2 (19-H), 18.0 (3-C), 16.4, 16.3, 16.2 (7, 11, 15-C); ν_max_/cm^–1^ 3265 (N-H), 1690 (C=O), 1653 (C=O), 1531 (N-H), 1223 (C-O); MS(ESI) m/z 425.1 (M-H)^-^; HRMS (Found (M + H)^+^ 427.2047. Calcd. for C_16_H_27_O_6_N_8_: (M + H)^+^ 427.2048.); (Purity test: HPLC method B).

### 3.3. Microbiology

All MICs were determined using an agar dilution method [41]. This necessitated the use of a defined antagonist-free medium (peptone-free), prepared as previously described with the inclusion of 2% saponin-lysed horse blood, 25 µg/mL NAD and 25 µg/mL haemin [23]. Test compounds were dissolved in sterile deionised water and incorporated into the antagonist-free medium at a final concentration range of 1–128 mg/L. All bacteria were prepared from fresh (18 h) subcultures on blood agar. Each isolate was suspended in sterile deionised water to a density equivalent to 0.5 McFarland units using a densitometer (approx. 1.5 × 10^8^ CFU/mL) and then diluted 1 in 15. A 1 µL aliquot of each diluted suspension was then delivered onto plates with a multipoint inoculator to give a final inoculum of approximately 10,000 CFU/spot as recommended [41]. Selected compounds of interest were tested without performing the 1 in 15 dilution resulting in a heavier inoculum of 150,000 CF/spot. This allowed us to assess their ability to inhibit large inocula of commensal bacteria that might potentially be present in clinical samples. All plates (including antimicrobial-free controls) were incubated for 22 h at 37 °C. The minimum inhibitory concentration (MIC) was recorded as the lowest concentration of test compound that resulted in complete inhibition of bacterial growth.

## 4. Conclusions

This paper introduces novel antibacterial agents specifically inhibiting the growth of certain bacteria. Such compounds can potentially be exploited to increase the selectivity and effectiveness of the detection of clinically important pathogens by chromogenic or fluorogenic culture media.

Antibacterial agents targeting the common bacterial enzyme alanine racemase were developed and investigated. Two 1-aminoalkyltetrazoles were synthesized as potential alanine racemase inhibitors. All molecules were tested against a collection of clinically relevant pathogenic bacteria. As the transportation of a single alanine unit into the cell is not provided by any bacteria, these molecules did not inhibit the growth of any bacteria when deployed as the simple amino acid analogue.

Selected di- and tri- and succinyl oligo-peptide analogues of l-1-aminoethyltetrazole were designed and synthesized by a novel solid phase peptide coupling method to enable the transportation of these molecules into specially targeted bacterial cells. Many of these peptides displayed significant antibacterial activity. Depending upon the attached amino acids, the substrates caused growth inhibition of bacteria with different levels of selectivity. The selective inhibition of enterococci while not affecting the growth of other Gram-positive cocci by l-pyroglutamyl dipeptide **14f-LL** is potentially a valuable result. The most significant example is the selective inhibition of *E. coli* and some *Enterobacterales* by l-alanyl- **14a-LL**, l-leucyl- **14h-LL** and dialanyl- **14c-LLL** substrates. These compounds should be investigated in clinical applications, as they could provide a selective antibacterial ingredient in a *Salmonella* selective culture medium.

In this report an important limiting factor is that very few results are presented to demonstrate inhibition of *E. coli* and growth of *Salmonella* in the presence of these inhibitors, as only two representative strains of each have been tested. However, the successful application of an analogous inhibitor, alafosfalin, in culture media for *Salmonella* gives us some expectation that these new inhibitors could demonstrate equivalent utility [42]. Further investigations are clearly warranted and these need to include the testing of a large collection of *Salmonella* isolates as well as large numbers of other species of *Enterobacterales*, using procedures that have previously been documented for analogous compounds such as alafosfalin [42]. Furthermore, for any new culture medium, multicentre trials with clinical specimens would be required, as previously described [42].

## 5. Patents

R. J. Anderson, M. Gray, L. Kondacs, S. Orenga, J. Perry, WO 2019/058074 A1 2019.

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
