# Peer review of "C-Terminal 1-Aminoethyltetrazole-Containing Oligopeptides as Novel Alanine Racemase Inhibitors"

_molecules, 2020, doi:10.3390/molecules25061315_

Round 1
Reviewer 1 Report
I have one question that might be helpful to be included in the introduction or discussion part. For the compound that partially inhibits some pathogens not not all, e.g. Salmonella, it is not really solid conclusion that this compound can be used as inhibitor during incubation stage to inhibit other pathogen growth. Because it is possible that that compound might also not inhibit some other strains that were not tested in this study. It is impossible to test all strains, especially considering many strains were not isolated in real application environment. So it will be helpful for readers to explain more that why this compound just does not inhibit Salmonella but will inhibit all other pathogens that have not been tested.
Author Response
Reviewer comment:
I have one question that might be helpful to be included in the introduction or discussion part. For the compound that partially inhibits some pathogens not all, e.g. Salmonella, it is not really solid conclusion that this compound can be used as inhibitor during incubation stage to inhibit other pathogen growth. Because it is possible that that compound might also not inhibit some other strains that were not tested in this study. It is impossible to test all strains, especially considering many strains were not isolated in real application environment. So it will be helpful for readers to explain more that why this compound just does not inhibit Salmonella but will inhibit all other pathogens that have not been tested.
Author response:
We accept this point entirely. In this preliminary study, we do not present sufficient data to convince the reader that any of the compounds have proven utility for application in culture media, as we have only tested a limited panel of bacterial strains. However, some of the results that are reported for these novel molecules are entirely consistent with those found for the analogous inhibitor alafosfalin and this compound has shown proven utility in culture media that have been designed for isolation of Salmonella. This provides some expectation that one or more of these new molecules may have similar utility. To provide proof of such utility will require testing of very large numbers of bacterial strains. An additional paragraph of text has been added to (1) highlight this important limitation of our work (2) indicate why we might expect the compounds to be successful and (3) indicate, by means of a new reference, exactly how further work would need to be taken forward.
The additional text has been added to lines 677-685: In this report an important limiting factor is that very few results are presented to demonstrate inhibition of E. coli and growth of Salmonella in the presence of these inhibitors, as only two representative strains of each have been tested. However, the successful application of an analogous inhibitor, alafosfalin, in culture media for Salmonella gives us some expectation that these new inhibitors could demonstrate equivalent utility [42]. Further investigations are clearly warranted and these need to include the testing of a large collection of Salmonella isolates as well as large numbers of other species of Enterobacterales, using procedures that have previously been documented for analogous compounds such as alafosfalin [42]. Furthermore, for any new culture medium, multicentre trials with clinical specimens would be required, as previously described [42].
The new corresponding reference is found on lines 793-794: [42] Perry, J.D.; Riley, G.; Gould, F.K.; Perez, J.M.; Boissier, E.; Ouedraogo, R.T.; Freydière, A.M. Alafosfalin as a selective agent for isolation of Salmonella from clinical samples. J. Clin. Microbiol. 2002, 40, 3913-3916.
Reviewer 2 Report
This interesting paper reports the synthesis of tetrazole-containing analogues of alafosfalin that have incorporated oligopeptide components to aid cellular uptake together with biological evaluation of the products. Some potentially useful compounds are made.
The paper is well presented with the work clearly put into perspective. The development of the synthetic work is well described. New compounds are properly characterised in the experimental and literature references are given for known compounds. The spectroscopic data that I checked would appear to have been accurately reported and were consistent with the assigned structures.
The paper itself is well presented with only the obvious slip up on page 4 to be corrected. Apart from this I would publish this paper as submitted with minimal editorial revision.
Author Response
The typographical issue kindly pointed out by this reviewer on Page 4 has been rectified.